# Human endothelial cells promote a human neural stem cell type B phenotype via Notch signaling

Brenda Gutierrez[1,2] ✉, Tzu Chia Liu [1,2], Carly Rodriguez[2], Oier Pastor-Alonso [3], Hannah Lambing[3], Mercedes F. Paredes [3] & Lisa A. Flanagan [1,2,4,5] ✉

Neural stem and progenitor cell (NSPC) and vessel-forming endothelial cell (EC) communication throughout development and adulthood is vital for normal brain function. However, much remains unclear regarding coordinated regulation of these cells, particularly in humans. We find that contact with hECs increases hNSPC type B cells, which are GFAP-expressing adult NSPCs in the subventricular zone (SVZ), leading to generation of a human type B single-cell RNA sequencing (scRNAseq) dataset. Differential gene expression demonstrates an increase in Notch downstream mediators in type B hNSPCs after hEC contact. Blocking hNSPC Notch signaling, and reducing hEC expression of the Notch ligand DLL4, abrogates the effect of hECs on type B hNSPCs. We identify S100A6 and LeX as human type B cell markers, and analysis of the postnatal human SVZ confirms co-expression of GFAP, SOX2, S100A6, LeX and PROM1 in type B cells. Sites of contact are identified between type B hNSPCs and vasculature in the SVZ, providing evidence of human type B cell contact with hECs in the postnatal human brain. Thus, hEC contact promotes human type B cells via Notch signaling and these cells are in contact in stem cell niches in the human brain.

Neural stem and progenitor cells (NSPCs) in the brain give rise to mature neurons, astrocytes, and oligodendrocytes and are often in close proximity to vasculature in the developing cortex[1–3] and adult neural stem cell niches[4–8]. While the effects of blood vessel endothelial cells (ECs) on NSPCs have been well studied in rodent systems, little is known about the interaction of human ECs (hECs) and human NSPCs (hNPSCs).

The nervous and vascular systems form in parallel during development, often sharing guidance cues for proper growth and patterning[9]. Vascularization of the developing rodent brain occurs via sprouting from the perineural vascular plexus and the periventricular plexus[10,11]. Vessel-forming ECs in the developing rodent brain play roles in NSPC differentiation[12] and neuronal migration[13]. In human

development, the neural tube is surrounded by dense connective tissue, called the meninx primitiva, containing a perineural vascular network[14–16]. As the neural tube thickens, the perineural vascular network sprouts via angiogenesis to supply oxygen and nutrients to the ventricular-subventricular proliferative germinal zone at gestational week (GW) 6–7 and the developing cortex at GW15-25[14,15]. In vitro, hNSPCs enhance hEC vessel formation, indicating functional relationships between these cell types[17]. The human ventricular zone/subventricular zone (VZ/SVZ) becomes highly vascularized, but how the hECs affect hNSPCs during human brain development is not well understood.

EC-NSPC communication is maintained in neural stem cell niches of the adult brain such as the hippocampal dentate gyrus subgranular

[1]Department of Anatomy & Neurobiology, University of California Irvine, Irvine, CA, USA. [2]Sue & Bill Gross Stem Cell Research Center, University of California Irvine, Irvine, CA, USA. [3]Neurology Department and Weill Institute for Neuroscience, University of California San Francisco, San Francisco, CA, USA. [4]Department of Biomedical Engineering, University of California Irvine, Irvine, CA, USA. [5]Department of Neurology, University of California Irvine, Irvine, CA, USA. ✉e-mail: bgutier4@uci.edu; lisa.flanagan@uci.edu

zone[18–20] and the SVZ adjacent to the lateral ventricles[21–23]. The SVZ is the largest adult neural stem cell niche, home to glial fibrillary acidic protein (GFAP)-expressing NSPCs termed type B cells that are multipotent and can self-renew[24,25]. In rodents, type B cells produce type C intermediate transit amplifying progenitors that give rise to type A neuroblasts that travel to the olfactory bulb via the rostral migratory stream[22,26–28]. ECs in the rodent SVZ are in direct contact with type B cells[29], placing them in a position to affect type B cells via secreted factors and cell-cell contact.

In rodents, a number of EC secreted factors regulate NSPCs, promoting stem cell maintenance[30,31], stimulating NSPC self-renewal, and regulating differentiation[32–34]. However, whether direct contact by ECs regulates NSPCs is less clear. Several lines of evidence suggest contact may be critical. The rodent SVZ has a unique modified blood-brain barrier containing vessel regions lacking coverage by pericytes and astrocytic endfeet, allowing for direct cell-cell contact between ECs and NSPCs[7]. Slowly dividing type B cells have been identified in close contact with blood vessels in the mouse SVZ[5]. When EC contact was disrupted, mouse type B cells moved away from blood vessels and increased proliferation, demonstrating that EC contact could play a role in regulating stem cells in the adult SVZ niche[5]. In addition, an in vitro study found mouse ECs use contact-mediated signaling to maintain mouse NSPC quiescence and induce a type B cell phenotype[35]. Whether hEC secreted factors or contact affect hNSPCs is largely unknown but important to study as human and rodent brain differ in development, structure, and function. Furthermore, there are clear molecular and structural differences between human and rodent NSPC niches.

A major difference between rodent and human adult NSPC niches is the unique hypocellular gap in the human SVZ separating the cell bodies of GFAP-expressing hNSPCs from the ependymal cells lining the lateral ventricle[36]. These GFAP-expressing SVZ cells were identified as hNSPCs with multipotent and self-renewal capabilities[37] similar to the type B cells observed in the rodent SVZ[38]. Another difference is that the human SVZ has fewer proliferating progenitors compared to the rodent or monkey SVZ[36,39,40]. In adult rodents, neuroblasts from the SVZ undergo chain migration via a rostral migratory stream to the olfactory bulb to generate new neurons that incorporate into circuitry. However, it remains controversial whether new neurogenesis or a similar rostral migratory stream from the SVZ persists normally in human adults[36,40–45]. Directly studying hNSPC behavior and cellular interactions are critical for understanding human brain function, particularly since there are many differences between human and rodent NSPC niches that mean rodent systems do not fully model humans.

In this study, we used in vitro human cell systems to investigate hEC and hNSPC interactions. We found that hECs have profound effects on hNSPC phenotype as revealed by immunostaining and single-cell RNA sequencing (scRNAseq). ScRNAseq data was used to identify Notch signaling as a potential cell-cell contact mediated mechanism by which hECs regulate hNSPCs and mechanistic studies tested the role of Notch in controlling hNSPC phenotype.

## Results

### hEC co-culture increases the percentage of GFAP+SOX2+ type B hNSPCs

To investigate the role of hECs in hNSPC regulation, we co-cultured human fetal brain-derived hNSPCs with human endothelial colony-forming cell-derived endothelial cells (hECs) and assessed cell phenotype by immunostaining after 5 days. We stained for multiple neural markers and identified hECs in co-culture via the cell surface marker CD31, also known as platelet and endothelial cell adhesion molecule 1 (PECAM1). Staining for neuronal markers microtubule associated protein 2 (MAP2) and doublecortin (DCX) showed little to no neuronal differentiation in co-cultures (Supplementary Fig. 1a). We assessed astrocytes and neural stem cells by staining for glial fibrillary acidic

protein (GFAP) and SRY-box transcription factor 2 (SOX2). Interestingly, co-cultures had clusters of hNSPCs co-expressing GFAP and SOX2, not seen in media control or with hEC conditioned media (CM) containing hEC secreted factors (Fig. 1a, Supplementary Fig. 2a). Since type B NSPCs in the adult SVZ express both GFAP and SOX2, we considered these GFAP+SOX2+ cells as putative type B hNSPCs. HNSPC-hEC co-cultures demonstrated a significant increase in the percentage of GFAP+SOX2+ costained type B hNSPCs as well as cells stained individually for either GFAP or SOX2 compared to media control and hEC CM (Fig. 1b). There was no significant difference in the percentage of differentiated astrocytes (GFAP+SOX2-) in co-culture compared to media control, but there was a slight increase in these cells in hEC CM (Fig. 1b). In addition, HEC CM slightly increased the percentage of GFAP+ cells compared to media control (Fig. 1b). The effect of hECs on hNSPC phenotype was not due to the hEC media utilized for co-cultures (Supplementary Fig. 1b, c).

Two types of hNSPC groupings were observed in co-culture: clusters adjacent to hECs and those above hECs (Supplementary Fig. 1d). HECs promoted an increase in GFAP+ hNSPCs compared to media control regardless of hNSPC position relative to hECs (Supplementary Fig. 1d). Increased GFAP+SOX2+ hNSPC percentage in co-cultures compared to media control was observed as early as 1 day after co-culture (Supplementary Fig. 1e). The number of GFAP+SOX2+ hNSPCs significantly increased in hEC co-culture, more than doubling, from day 1 to day 5 (Supplementary Fig. 1f). In addition, hECs increased the percentage of GFAP+SOX2+ type B hNSPCs at various hNSPC to hEC ratios ranging from 1:1 to 1:5 (Supplementary Fig. 1g), demonstrating the effect of hECs on hNSPCs is consistent across different conditions.

To test whether the increase in GFAP+SOX2+ hNSPCs in hEC co-culture compared to media control was reproducible across different sets of cells, we co-cultured hECs with additional sets of human fetal brain-derived hNSPCs and saw a significant increase in the percentage of GFAP+SOX2+ type B hNSPCs in hEC co-culture compared to media control (Fig. 1c, Supplementary Fig. 2b, c). Another type of EC, human brain microvascular endothelial cells (hBMECs) co-cultured with hNSPCs also promoted an increase in the percentage of GFAP+SOX2+ type B hNSPCs compared to media control (Fig. 1d, Supplementary Fig. 1h, i, Supplementary Fig. 2d). The clustering of hNSPCs occurs with hEC and hBMEC co-culture, but hNSPCs were scattered when co-cultured with normal human lung fibroblasts (NHLFs) (Fig. 1e, Supplementary Fig. 2e). In addition, NHLFs did not stimulate an increase in the percentage of GFAP+SOX2+ hNSPCs after co-culture as seen with hECs (Fig. 1e and Supplementary Fig. 1j). In summary, the increase in GFAP+SOX2+ hNSPC percentage is reproducible using different hNSPC and hEC cells and is specific to hEC co-cultures.

### GFAP+ hNSPCs in hEC co-culture express additional type B NSPC markers LewisX and S100A6

To further characterize the GFAP+SOX2+ putative human type B hNSPCs, we stained for two additional type B cell markers: LewisX (LeX), also known as CD15 or stage-specific embryonic antigen 1 (SSEA-1), and S100 calcium binding protein A6 (S100A6). LeX is a carbohydrate expressed by type B cells in the rodent SVZ and has been used as a surface marker for fluorescence-activated cell sorting of quiescent adult mouse NSPCs[46,47]. To confirm LeX is associated with hNSPCs rather than differentiated cells, hNSPCs were differentiated and stained for LeX, revealing a significant decrease in the percentage of LeX+ cells after differentiation (Fig. 2a). Since LeX (hNSPCs) and CD31 (hECs) are cell surface markers, we stained cells in co-culture and used confocal imaging to observe points of contact between hNSPCs and hECs (Fig. 2b). HNSPC-hEC co-cultures stained for GFAP and LeX demonstrated a significant increase in the percentage of GFAP+LeX+ and LeX+ hNSPCs compared to media control and hEC CM (Fig. 2c, d, Supplementary Fig. 3a). Interestingly, there was a large increase in LeX

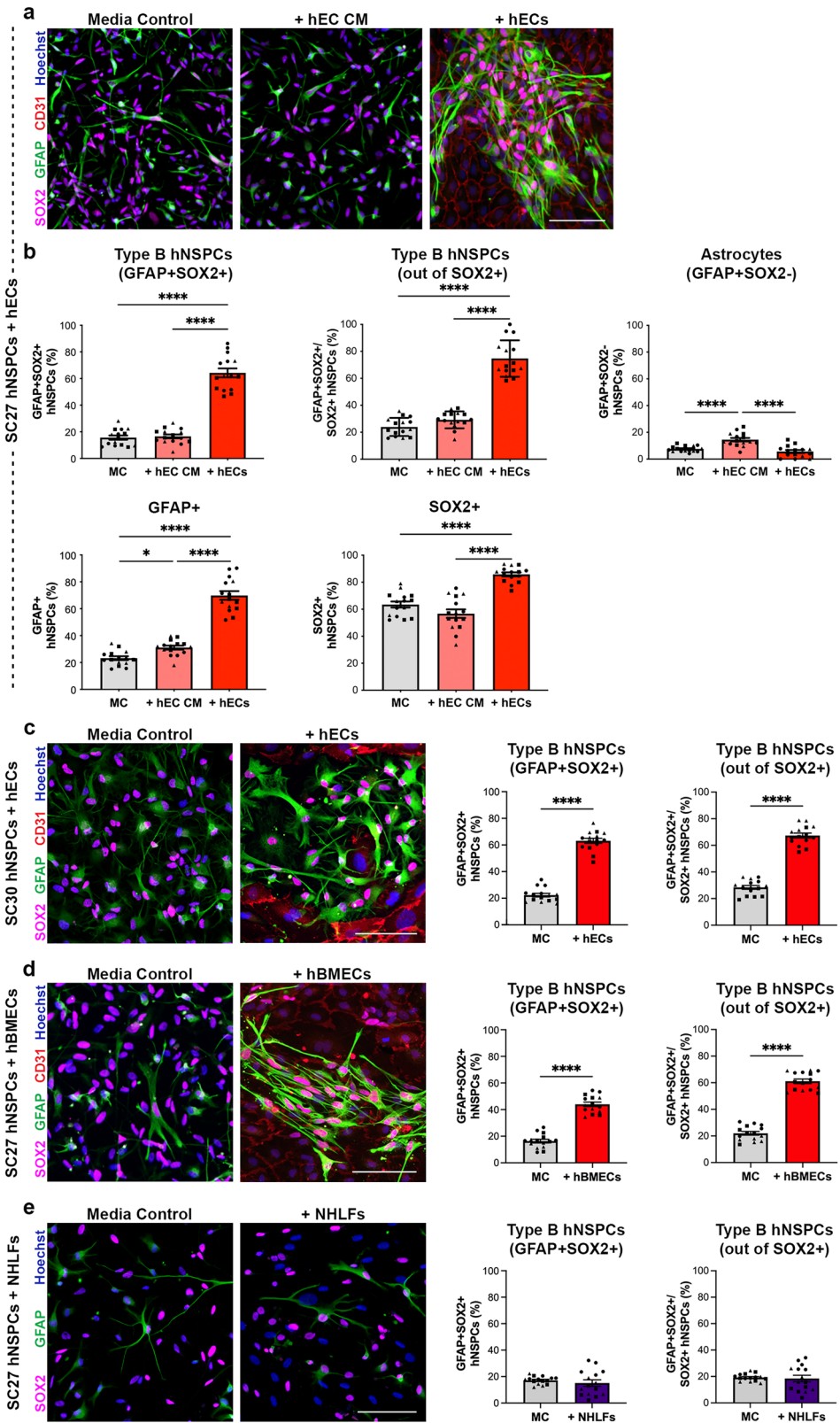

+ cells co-expressing GFAP after co-culture (83.35 ± 7.25%) compared to media control (13.47 ± 3.61%) or hEC CM (5.00 ± 3.62%) (Fig. 2d).

To better approximate the three-dimensional (3D) in vivo environment, we co-cultured hNSPCs and hECs in scaffolds mimicking human brain mechanical and extracellular matrix properties and stained for type B cell markers GFAP and LeX[17]. These markers were chosen since the antibodies were compatible with staining in 3D. As

observed in 2D environments, co-culture of hNSPCs with hECs in 3D led to increased co-expression of type B cell markers (Supplementary Fig. 4).

S100A6, involved in various cellular processes such as proliferation and differentiation, is a marker expressed by adult NSPCs in the rodent SVZ[48,49] and hippocampus[50]. HNSPCs differentiated for 5 and 21 days demonstrated a significant reduction in S100A6+ hNSPCs

**Fig. 1 | hECs increase the percentage of GFAP+SOX2+ human type B cells. a** SC27 hNSPCs in media control (MC), hEC conditioned media (hEC CM), or co-culture with hECs (+hECs) immunostained for GFAP, SOX2, and EC marker CD31. **b** GFAP+SOX2+ type B cells are significantly greater in co-culture with hECs compared to MC (****p < 0.0001) or hEC CM (****p < 0.0001). There is no significant difference in type B cell percentage between MC and hEC CM (p = 0.9532). Type B hNSPCs out of total number of SOX2+ hNSPCs are significantly greater in co-culture with hECs compared to MC (****p < 0.0001) or hEC CM (****p < 0.0001). There is no significant difference in the percentage of type B hNSPCs out of SOX2+ hNSPCs between MC and hEC CM (p = 0.2898). Astrocytes are higher in hEC CM compared to MC (****p < 0.0001) or hECs (****p < 0.0001), while there is no significant difference between MC and +hECs (p = 0.3840). GFAP+ cells and SOX2+ cells are highest in hECs co-culture compared to MC (****p < 0.0001 for GFAP+ and SOX2+) or hEC CM (****p < 0.0001 for GFAP+ and SOX2+). There is a significant increase in GFAP+ cells in hEC CM compared to MC (*p = 0.0384) and no significant difference in SOX2+ cells observed in hEC CM compared to MC (p = 0.1279). **c** SC30 hNSPCs in MC or +

hECs immunostained for GFAP, SOX2, and CD31. The percentage of GFAP+SOX2+ type B hNSPCs in hEC co-culture significantly increased compared to MC whether out of total hNSPCs (****p < 0.0001) or SOX2+ hNSPCs (****p < 0.0001). **d** SC27 hNSPCs in MC or with hBMEC co-cultures (+hBMECs) immunostained for GFAP, SOX2, and CD31. The percentage of GFAP+SOX2+ type B hNSPCs was significantly higher in +hBMECs compared to MC, whether out of total hNSPCs (****p < 0.0001) or SOX2+ hNSPCs (****p < 0.0001). **(e)** SC27 hNSPCs in MC or NHLF co-culture (+NHLFs) immunostained for GFAP and SOX2. The percentage of GFAP+SOX2+ type B hNSPCs was not significantly different between MC and +NHLFs, whether out of total hNSPCs (p = 0.4191) or SOX2+ hNSPCs (p = 0.7655). All images represent day 5 co-cultures. Nuclei stained with Hoechst. Scale bars, 100 μm, n = 3 independent biological replicates indicated by symbols, 5 areas quantified per sample, percentages are out of total hNSPCs unless otherwise noted, mean with SEM. Analysis used (**b**) one-way ANOVA with Tukey post-hoc test for multiple comparisons and (**c–e**) unpaired two-tailed Student's t-test. Source data are provided as a Source Data file.

compared to undifferentiated hNSPCs (Fig. 2e). Staining hNSPC-hEC co-cultures for GFAP and S100A6 revealed a significant increase in the percentage of GFAP+S100A6+ type B hNSPCs compared to media control but no significant difference when compared to hEC CM (Fig. 2f, g, Supplementary Fig. 3b). The percentage of GFAP+S100A6+ out of S100A6+ hNSPCs was significantly higher in hEC co-culture compared to media control and hEC CM (Fig. 2g). There was no significant difference in the percentage of S100A6+ hNSPCs in co-culture versus media control and hEC CM (Fig. 2g). Similar increases in GFAP+S100A6+ cells after hEC co-culture were observed with a second set of hNSPCs (Supplementary Fig. 3c). Overall, hEC co-culture consistently increased the percentage of type B hNSPCs identified via expression of GFAP, SOX2, LeX, and S100A6.

### The effect of hEC secreted factors on hNSPCs varies

HEC CM did not significantly increase the percentage of GFAP+SOX2+ (Fig. 1b), GFAP+LeX+ (Fig. 2d), and GFAP+S100A6+ (Fig. 2g) type B hNSPCs compared to media control. To further explore the role of hEC secreted factors on hNSPC phenotype, we conducted a transwell study to capture hEC secreted factors with short half-lives that could have been missed in the initial CM study (Supplementary Fig. 5a). HECs were plated in the insert of the transwell and SC27 hNSPCs in the bottom compartment. There was no effect of hEC secreted factors on the percentage of GFAP+SOX2+ type B hNSPCs, GFAP+SOX2- astrocytes, GFAP+ hNSPCs, and SOX2+ hNSPCs compared to media control or with hNSPCs in the transwell insert as a control (Supplementary Fig. 5a, b).

We tested the potential role of hEC secreted factors using additional cell sources. HEC CM significantly increased the percentage of GFAP+SOX2+SC30 hNSPCs compared to media control but not to the same degree as hEC co-culture (Supplementary Fig. 6a, b). In addition, hBMEC CM stimulated a significant increase in GFAP+SOX2+ (Supplementary Fig. 6c, d, e) and GFAP+LeX+ (Supplementary Fig. 6f, g, h) type B hNSPCs (SC27) compared to media control but not to the same degree as hBMEC co-culture. The hBMEC CM effect on GFAP+SOX2+ and GFAP+LeX+ percentage decreased with hBMEC passage so that by the third passage, the CM treated cells were similar to media control (Supplementary Fig. 6e, h). In summary, while there may be a role of hEC secreted factors on hNSPC phenotype, this effect was not consistent across different cell sources. In contrast, hEC co-culture demonstrated reproducible results across different hNSPC and hEC sources. We therefore focused on hEC co-cultures and contact-mediated mechanisms of type B cell formation.

### Single cell RNA sequencing demonstrates GFAP-expressing cells are human type B cells

To further characterize GFAP+SOX2+ cells, we conducted single cell RNA sequencing (scRNAseq) of hNSPC-hEC co-cultures and hNSPC

controls. For cluster identification, we merged samples and performed uniform manifold approximation and projection (UMAP) analysis for dimension reduction after quality control (Fig. 3a and Supplementary Fig. 7a, b). We analyzed the following number of cells per condition: media control: 0 hECs, 10,780 hNSPCs; conditioned media: 0 hECs, 13,871 hNSPCs; co-culture 7,375 hECs, 158 hNSPCs. The lower number of hNSPCs in the co-culture condition is due to the 1:5 ratio of hNSPCs:hECs, the higher proliferation rate of hECs compared to hNSPCs, and the increased adhesion of hNSPCs to the tissue culture plate after co-culture making dissociation challenging. The lower number of cells does not significantly affect cluster identification as merged datasets were used to identify clusters and the number of cells is sufficient to determine the proportion in different clusters. The groups identified were endothelial cells (with a small subset actively proliferating), neural stem cells, transitional cells, astrocyte progenitors, and proliferating progenitors (Supplementary Fig. 7b and Supplementary Data 1). PECAM1 (CD31) was expressed by hEC clusters, SOX2 was expressed by hNSPC clusters, and both cell types had proliferating clusters (Fig. 3b). A subset of neural stem cells expressed GFAP and SOX2 (Fig. 3b) and we performed subset analysis of just the hNSPC clusters to further distinguish distinct hNSPC cell types.

Subset analysis identified 6 distinct hNSPC populations: neural stem cells, transitional cells, type B cells, astrocyte progenitors, proliferating progenitors, and early neuron progenitors (Fig. 3c and Supplementary Data 2). Type B cells expressed several interesting genes in addition to GFAP, including the stem cell marker PROM1 (CD133)[51–53] and HES1, a Notch downstream mediator known to be involved in the maintenance of NSPCs[54,55] (Fig. 3d). Type B cells expressed NGFR (CD271), which is a marker of hNSPCs in the adult human SVZ[56,57], and apolipoprotein E (APOE) that is expressed by adult rodent hippocampal NSPCs where it plays a role in NSPC maintenance[58]. Type B hNSPCs and neural stem cells expressed SLC1A3, which is the glutamate transporter GLAST that is associated with rodent fetal and adult NSPCs as well as astrocytes[59] (Fig. 3d). S100A6 was expressed by type B hNSPCs and was also highly expressed in the neural stem cell cluster (Fig. 3d). Although SOX2 is expressed by all hNSPC clusters (Fig. 3b), the highest expression was by proliferating progenitors (Fig. 3d). Interestingly, genes that negatively regulate cell proliferation such as tumor protein p53 inducible protein 11 (TP53I11) and BTG anti-proliferation factor 1 (BTG1), were among the top genes in the type B hNSPC cluster (Fig. 3e and Supplementary Data 2). Cell cycle analysis using Seurat confirmed type B hNSPCs were not in the S, G2, or M phases of the cell cycle, suggesting low proliferation (Supplementary Fig. 7c).

Transitional cells had low expression of type B cell markers and proliferation markers MKI67 and TOP2A, suggesting they may be transitioning between type B cells and proliferating progenitors (Fig. 3d). Proliferating progenitors expressed proliferation markers

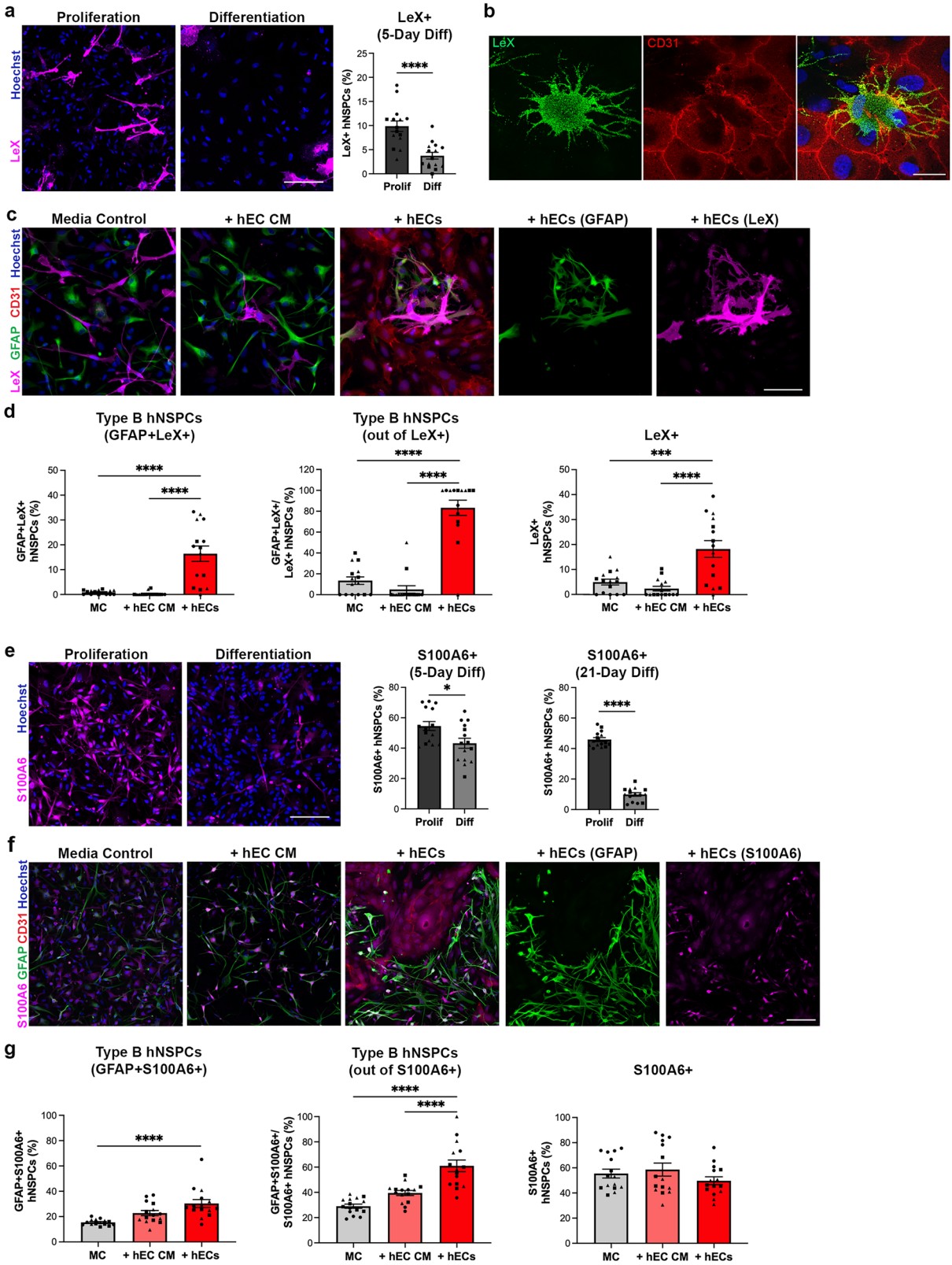

along with low levels of astrocyte progenitor markers FABP7 and SPARCL1, making it possible they are proliferating astrocyte progenitors (Fig. 3d). In addition to FABP7 and SPARCL1, astrocyte progenitors expressed the astrocyte marker aquaporin 4 (AQP4) (Fig. 3d). Lastly, we identified a small cluster of early neuron progenitors expressing the pro-neural genes ASCL1, DCX-like kinase 2 (DCLK2), and MAP2[60] (Fig. 3d).

## Human type B cluster is more like in vivo rodent and human type B cells than radial glia

A difficulty with cluster identification is the overlap among markers for type B cells and radial glial cells (RGCs), which are the NSPCs of the developing brain that give rise to adult type B cells[25,59,61,62]. We used published rodent scRNAseq data from in vivo cells to help us identify markers useful for human type B cell cluster identification. We first

**Fig. 2 | hEC co-culture increases the percentage of GFAP+LeX+ and GFAP +S100A6+ human type B cells. a** SC27 hNSPCs differentiated (Diff) for 5 days and immunostained for LeX demonstrate a significant decrease in the percentage of LeX+ hNSPCs compared to undifferentiated hNSPCs in proliferation media (Prolif) (****$p < 0.0001$). **b** Co-culture of SC27 hNSPCs with hECs and immunostaining for cell surface markers LeX for hNSPCs and CD31 for hECs show cell contact in confocal imaging (scale bar 30 μm). **c** SC27 hNSPCs in MC, hEC CM, or with hECs stained for GFAP and LeX. **d** GFAP+LeX+ type B hNSPCs are significantly greater in co-culture with hECs compared to MC (****$p < 0.0001$) or hEC CM (****$p < 0.0001$). Type B hNSPCs out of total number of LeX+ hNSPCs are significantly greater in co-culture with hECs compared to MC (****$p < 0.0001$) or hEC CM (****$p < 0.0001$). LeX+ hNSPCs are significantly greater in co-culture with hECs compared to MC (***$p = 0.0002$) or hEC CM (****$p < 0.0001$). There was no significant difference between hEC CM and MC for GFAP+LeX+ ($p = 0.971$), GFAP+LeX+ out of LeX+ ($p = 0.478$), and LeX+ hNSPCs ($p = 0.6614$). **e** HNSPCs differentiated for 21 days and immunostained for S100A6. The percentage of S100A6+ hNSPCs significantly decreased for 5-day (*$p = 0.0154$) and 21-day (****$p < 0.0001$) differentiated hNSPCs compared to undifferentiated hNSPCs. **f** SC27 hNSPCs in MC, hEC CM, or + hECs immunostained for GFAP and S100A6. **g** The percentage of GFAP+S100A6+ type B hNSPCs is significantly greater in hEC co-culture compared to MC (****$p < 0.0001$). The percentage of GFAP+S100A6+ hNSPCs out of S100A6+ hNSPCs is significantly greater in hEC co-culture compared to MC (****$p < 0.0001$) and +hEC CM (****$p < 0.0001$). The percentage of S100A6+ hNSPCs does not significantly differ in hEC co-culture compared to MC ($p = 0.5816$) and +hEC CM ($0.2749$). There was no significant difference between hEC CM and MC for GFAP+S100A6+ ($p = 0.0586$), GFAP+S100A6+ out of S100A6+ ($p = 0.0518$), and S100A6+ hNSPCs ($p = 0.8424$). Co-cultures were grown for 5 days. Nuclei were stained with Hoechst. Scale bars, 100 μm unless otherwise noted, $n = 3$ independent biological replicates indicated by symbols, 5 areas quantified per sample, percentages are out of total hNSPCs unless otherwise noted, mean with SEM. Analysis used (**a, d**) unpaired two-tailed Student's t-test and (**c, f**) one-way ANOVA with Tukey post-hoc test for multiple comparisons. Source data are provided as a Source Data file.

integrated mouse RGC clusters from the Li et al. study[63] with the mouse quiescent type B cell cluster from Cebrian-Silla et al.[64] using CCA integration in Seurat. UMAP dimensionality reduction of integrated data was conducted and RGCs clustered well with quiescent type B cells (Fig. 4a).

We used differential gene expression analysis of mouse type B cells versus mouse RGCs to clarify which genes are up-regulated in type B cells. This analysis revealed multiple genes highly expressed by mouse type B cells (Fig. 4b, Supplementary Data 3). Cells in our putative hNSPC type B cell cluster expressed several of the same genes, including GFAP, S100A6, MALAT1, APOE, BTG1, NRXN3, MT3, NTRK2, TIAM2, and CEBPD (Fig. 4b, Supplementary Data 3). In fact, 13 of the top 50 human type B cell genes were associated with mouse type B cells while only 2 of the 50 were genes up-regulated in mouse RGCs (Fig. 4c, Supplementary Data 3). In order to determine gene expression in type B cells compared to RGCs, we used the UCSC Cell Browser[65] and published gene lists to examine the expression of identified genes in 3 human RGC scRNAseq datatsets[66–68] as well as 1 additional mouse RGC[69] and 4 additional mouse type B cell datasets[70–73]. GFAP, APOE, MT3, and NTRK2 were commonly expressed by type B cells and astrocytes, while S100A6 was more specific to type B cells. Borrett et al., also found Gfap, S100a6, Apoe, Mt3, and Ntrk2 were up-regulated in rodent quiescent type B cells compared to RGCs and/or type C transit amplifying progenitors[73]. Expression of S100a6, Mt3, Apoe, Cebpd, Gfap, Junb, Ramp1, and Itm2b is higher in mouse type B cells, while Srsf6, Map1b, Tia1, Sox11, Foxg1, Matr3, and Dhx9 expression is higher in mouse RGCs (Fig. 4d, Supplementary Fig. 8a, 9a, Supplementary Data 3).

To determine whether these markers hold true for human in vivo RGCs and type B cells, we integrated human RGCs from Nowakowski et al.[66], adult SVZ neural stem cell-like cells identified by Baig et al.[74], and our putative type B hNSPCs. CCA integration was performed and UMAP was used for visualization (Fig. 4e). As mouse type B cells are adult SVZ NSPCs, we renamed Baig et al.'s neural stem cell-like cells as type B cells for nomenclature consistency between species. As with the mouse dataset, human type B cells, including our type B hNSPCs, demonstrate higher expression of S100A6, MT3, APOE, CEBPD, GFAP, JUNB, RAMP1, and ITM2B while human RGCs more highly express SRSF6, MAP1B, TIA1, SOX11, FOXG1, MATR3, and DHX9 (Fig. 4f, Supplementary Fig. 8b, 9b, Supplementary Data 4).

Baig et al. identified subclusters within their type B hNSPCs[74]. To determine whether our type B hNSPC cluster contained subclusters as well, we performed subset analysis of type B hNSPCs and found two type B hNSPC clusters (Supplementary Fig. 10a). Cells in type B subcluster 1 highly expressed S100A6 along with STXBP6, NEFL, and NGFR (Supplementary Fig. 10b, Supplementary Data 5). Cells in the smaller type B subcluster 2 demonstrated higher expression of COL1A2, NEAT1, IER3, and PCDH9 (Supplementary Fig. 10b, Supplementary

Data 5). Analysis of subcluster specific genes and GFAP revealed varying expression patterns across the clusters and degree of co-expression (Supplementary Fig. 10c). Of the genes analyzed, S100A6 was co-expressed with GFAP in 86.08% of type B hNSPCs, which was higher than the other markers examined (Supplementary Fig. 10c, d).

## hEC co-culture increases type B hNSPCs while decreasing progenitors

After cluster identification, we examined changes in the hNSPC population composition in media control compared to hEC co-culture (Supplementary Fig. 11a). No novel hNSPC clusters were identified in hEC co-culture, but the scRNAseq data showed a shift in the percentage of cells found in various clusters (Fig. 5a). HNSPCs in hEC co-culture contained 56.3% type B cells compared to 30.2% in media control (Fig. 5a), coinciding with our immunostaining results (Figs. 1, 2). Unexpectedly, there was a decrease in the percentage of astrocyte progenitors and proliferating progenitors in co-culture compared to media control (astrocyte progenitors: co-culture 13.3%, and media control 26.0%; proliferating progenitors: co-culture 3.8% vs. media control 17.4%).

HEC CM increased the percentage of type B hNSPCs (41.4%) compared to media control but not to the same extent as hEC co-culture and at the expense of neural stem cells as hEC CM had a lower percentage of neural stem cells (2.5%) compared to media control (16.7%) and hEC co-culture (20.3%) (Supplementary Fig. 11b). Grouping together neural stem cells and type B neural stem cells, media control had a total of 46.9% and hEC CM had 43.9%, while hEC co-culture had 76.6%. Unlike hEC co-culture, hEC CM did not change astrocyte progenitor or proliferating progenitor percentages compared to media control.

To confirm the decrease in astrocyte progenitors in co-culture, we differentiated hNSPCs after hEC co-culture and measured astrocyte formation. HEC viability is low in hNSPC differentiation media so we developed a differential dissociation protocol to remove hECs from co-culture, taking advantage of their lower adhesion to the laminin-coated surface. After dissociating hECs in co-culture, the remaining hNSPCs were replated on coverslips, grown in proliferation media for one day, then switched to astrocyte differentiation media for 5 days (Fig. 5b). Differentiated hNSPCs were stained for the astrocyte marker AQP4 as well as CD31 to ensure successful removal of hECs (Fig. 5c). HNSPCs previously in co-culture demonstrated a decrease in the percentage of AQP4+ astrocytes by approximately half compared to hNSPCs in media control (media control: $12.93 \pm 0.69$, +hECs: $6.6 \pm 0.56$) (Fig. 5c). This coincides with scRNAseq results demonstrating hEC co-culture reduced the astrocyte progenitor population by half (Fig. 5a), and the presence of only weakly AQP4-positive cells in co-culture conditions is also consistent with lowered astrocyte differentiation (Fig. 5c). Therefore, hNSPCs co-cultured with hECs had fewer

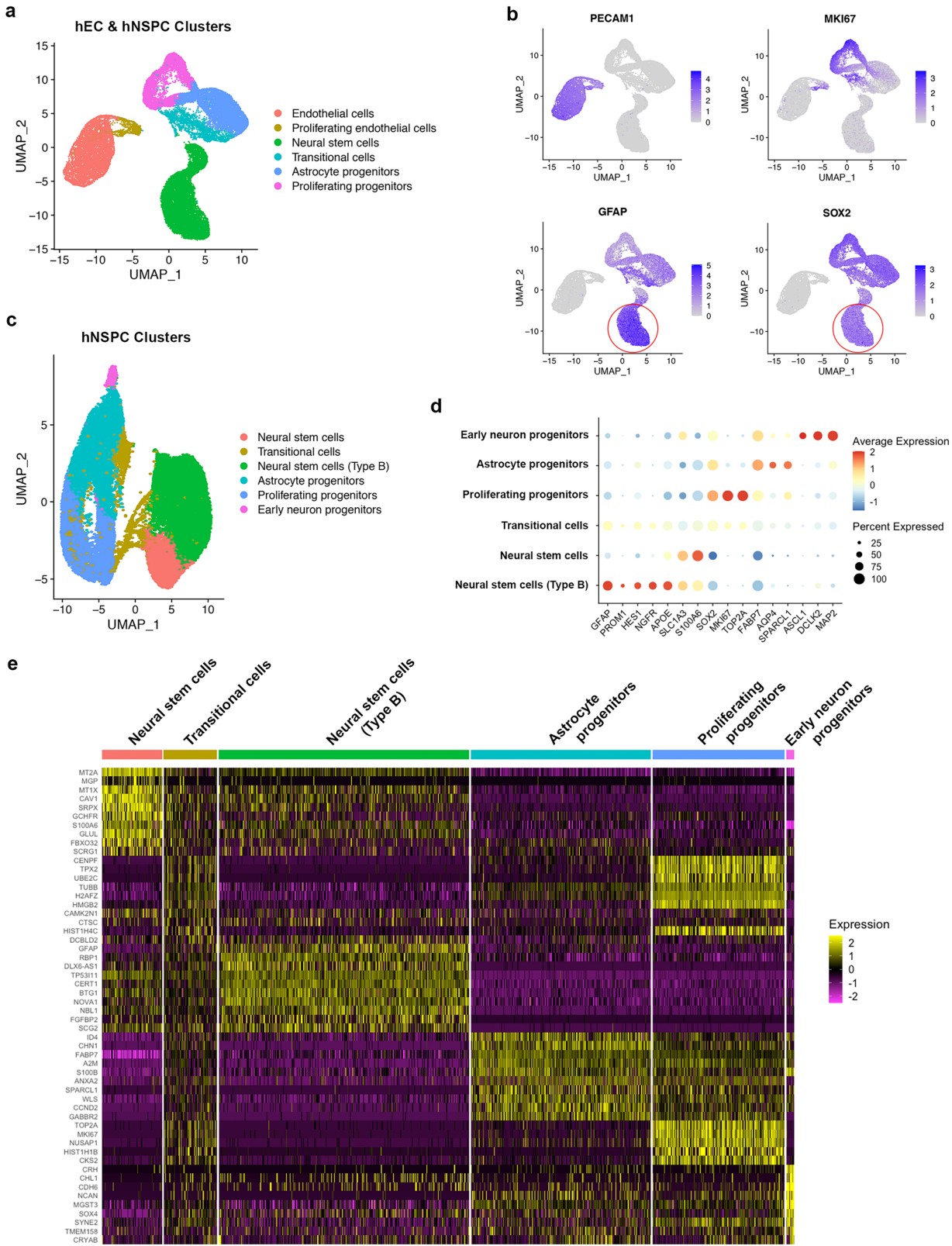

astrocyte progenitors identified by scRNAseq and diminished astro-cytic differentiation.

Since there was a decrease in proliferating progenitors detected by scRNAseq, we investigated the effect of hECs on hNSPC prolifera-tion. Co-cultures were stained for Ki67 or incubated with EdU to label proliferating cells (Fig. 5d, e). We found a significant reduction in the percentage of Ki67+ hNSPCs and EdU+ hNSPCs in hEC co-culture

compared to media control (Fig. 5d, e). In contrast, hEC CM did not affect the percentage of Ki67+ and EdU+ hNSPCs compared to media control (Supplementary Fig. 11c, d) and the percentage of EdU+ hNSPCs did not change after co-culture with NHLFs (Supplementary Fig. 11e). Comparing the growth of hNSPCs over time, the total number of hNSPCs more than tripled in media control on day 5 compared to day 1, while the number of hNSPCs in hEC co-culture merely doubled

**Fig. 3 | GFAP-positive hNSPCs express type B cell markers. a** UMAP plot of merged scRNAseq datasets of hNSPCs in media control, hEC conditioned media, and hEC co-cultures. **b** Featureplots demonstrate PECAM1 (CD31) expression by hEC clusters, MKI67 expression by proliferating endothelial cells and proliferating progenitors, and GFAP and SOX2 expression by hNSPC clusters. High GFAP and SOX2 co-expression was observed in a subset of neural stem cells (red circles). **c** UMAP plot of hNSPC subsets from merged datasets revealed two additional clusters, neural stem cells (Type B) and early neuron progenitors. **d** Dot plot

demonstrating the expression of type B cell markers, neural stem cell markers, proliferation markers, astrocyte markers, and neuronal markers. The average gene expression of cells in the cluster compared to the average expression across all clusters is noted by the color. The size of the dot represents the percentage of cells in the cluster expressing a particular gene. **e** Heatmap of the top 10 differentially expressed genes in each hNSPC cluster depicted in (**c**) based on average log2 fold change. Yellow represents high expression and purple is low expression.

(Fig. 5f). This difference was not due to increased cell death as there was little to no cleaved caspase 3 staining in co-culture and media control groups (Supplementary Fig. 11f), indicating lower hNSPC proliferation as the primary reason for reduced cell numbers in hEC co-culture.

### Up-regulation of genes related to adhesion and Notch signaling in type B hNSPCs after hEC contact

To identify cellular changes in type B hNSPCs in co-culture versus media control, we conducted differential gene expression analysis using Seurat (Supplementary Data 6). Multiple genes were up-regulated in type B hNSPCs after co-culture with hECs (Fig. 6a). Enrichr, a gene list enrichment analysis tool, was utilized to run Gene Ontology (GO) cellular component analysis of genes up-regulated in co-culture type B hNSPCs[75–77]. Up-regulated genes were involved in collagen-containing extracellular matrix, focal adhesions, and cell-substrate junctions (Fig. 6b, Supplementary Data 7). Interestingly, we noted increased hNSPC adhesion to laminin-coated plates after co-culture with hECs demonstrated by a significant increase in the number of hNSPCs remaining adhered after dissociation of co-cultures compared to the media control (Fig. 6c, d).

To explore mechanisms by which hECs affect hNSPC phenotype, we used the Panther metabolic and cell signaling pathway database in Enrichr and found activation of Notch signaling pathway in co-cultures (Fig. 6e, Supplementary Data 7). The Notch pathway is known to promote quiescence and maintenance of NSPCs in the embryonic and adult mouse brain[35,78–80]. Notch signaling is a contact-mediated mechanism whereby binding of Notch ligands, such as DLL and JAG, to Notch receptors leads to the cleavage of the Notch intracellular domain by γ-secretase. The Notch intracellular domain then translocates to the nucleus where it aids in the transcription of Notch downstream mediators such as the HES and HEY transcription factors. HES and HEY are involved in hNSPC quiescence and maintenance by inhibiting the expression of proneural genes[81]. Differential gene expression analysis of the type B hNSPC clusters in hEC co-culture compared to media control identified up-regulation of HES4 and HEY1, not observed with hEC CM (Fig. 6a and Supplementary Fig. 12). ScRNAseq data indicated significant Notch signaling in co-cultures, suggesting hECs may activate Notch signaling in hNSPCs.

### Notch signaling inhibition reduces the percentage of GFAP +SOX2+ type B hNSPCs

To determine whether Notch signaling is involved in increasing the percentage of type B hNSPCs in hEC co-culture, we inhibited Notch signaling using DAPT, a γ-secretase inhibitor that prevents the cleavage of the Notch intracellular domain. DAPT concentration of 10 µM was sufficient to decrease HEY1 and HES4 in hNSPCs (Supplementary Fig. 13a). Treatment of co-cultures with DAPT increased the number of hECs (Supplementary Fig. 13b), which could be due to the fact that Notch signaling suppresses endothelial cell proliferation[82]. To avoid effects of DAPT on hECs, we disrupted Notch only in hNSPCs by pre-treating hNSPCs with DAPT before plating with hECs (Fig. 6f). HNSPC pre-treatment with DAPT significantly decreased hNSPC HEY1 expression and the reduction was maintained for 24 h after DAPT was removed from the media (Fig. 6g and Supplementary Fig. 13c). Since 1 day of co-culture is sufficient for the hEC-induced increase in

GFAP+SOX2+ type B hNSPCs (Supplementary Fig. 1e), we co-cultured DAPT pre-treated hNSPCs with hECs for 1 day. HEY1 reduction was evident in co-cultures containing DAPT pre-treated hNSPCs but not in those with DMSO pre-treated hNSPCs (Fig. 6h). In addition, the expression of GFAP and tenascin C (TNC), an up-regulated gene after hEC co-culture (Supplementary Data 6), was reduced in co-cultures with DAPT pre-treated hNSPCs compared to the DMSO control while SOX2 expression did not significantly change (Fig. 6h).

Staining of co-cultures for GFAP, SOX2, and CD31 (Fig. 6i) revealed a significant decrease in the percentage of GFAP+SOX2+ type B hNSPCs in DAPT pre-treated hNSPC co-cultures compared to DMSO controls (Fig. 6j). DAPT treatment did not decrease the percentage of GFAP+SOX2+ type B hNSPCs in monoculture but blocked the increase in type B cells stimulated by hECs in co-culture, suggesting that Notch is critical for the hEC effect (Fig. 6j). The decrease in GFAP+SOX2+ hNSPCs in DAPT pre-treated co-cultures compared to DMSO controls is driven by a significant decrease in GFAP+ hNSPCs as the percentage of SOX2+ hNSPCs did not change. This may be due to the shorter 1-day co-culture time in these experiments since SOX2 expression decreases slowly in hNSPCs; even after 5 days of differentiation there is only approximately a 30% reduction in SOX2 expression and roughly 60% of hNSPCs still express SOX2 (Supplementary Fig. 13d). These results show that blocking Notch signaling in hNSPCs prevents the hEC-mediated increase in type B hNSPCs.

### hEC DLL4 knock-down diminishes the increase of type B hNSPCs in co-culture

To determine potential Notch ligand and receptor pairs, we first examined the expression of Notch ligands by hECs and found hECs express DLL4, JAG1, and JAG2. Both DLL4 and JAG2 are expressed by hECs but not hNSPCs (Fig. 7a, Supplementary Fig. 14a). DLL4 is primarily expressed by the larger cluster of non-proliferative hECs rather than the small cluster of proliferative cells (Figs. 3a, 7a). HNSPCs expressed the Notch receptors NOTCH1, NOTCH2, and NOTCH3 (Fig. 7b). To investigate which Notch ligands presented by hECs promote an increase in type B hNSPCs, we plated hNSPCs on coverslips coated with recombinant human Notch ligand Fc chimera proteins DLL4, JAG1, and JAG2. Notch ligands increased the expression of HEY1 and HES4 by hNSPCs thus confirming Notch activation (Fig. 7c, Supplementary Fig. 14b). DLL4 was the only ligand that increased hNSPC expression of GFAP, SOX2, and TNC (Fig. 7c, Supplementary Fig. 14b). Therefore, we studied the effect of hEC DLL4 on type B hNSPCs by reducing hEC expression of DLL4 using siRNA (Fig. 7d). HECs incubated with DLL4 siRNA or non-targeting (NT) siRNA were plated in co-culture with hNSPCs and stained for GFAP, SOX2, and CD31 (Fig. 7e). We found a significant decrease in the percentage of type B hNSPCs in co-culture after hEC DLL4 knock-down compared to the NT control and no significant difference from media control (Fig. 7f). The NT control is critical since treatment with siRNA may unexpectedly impact type B cell formation. HECs with reduced expression of DLL4 significantly decreased the percentage of GFAP+ hNSPCs compared to the NT control, while SOX2 trended toward a decrease in percentage but was not statistically significant (Fig. 7f). This corroborates the DAPT experiment where GFAP expression was significantly reduced while SOX2 trended toward a decrease in expression (Fig. 6h). These results indicate hEC

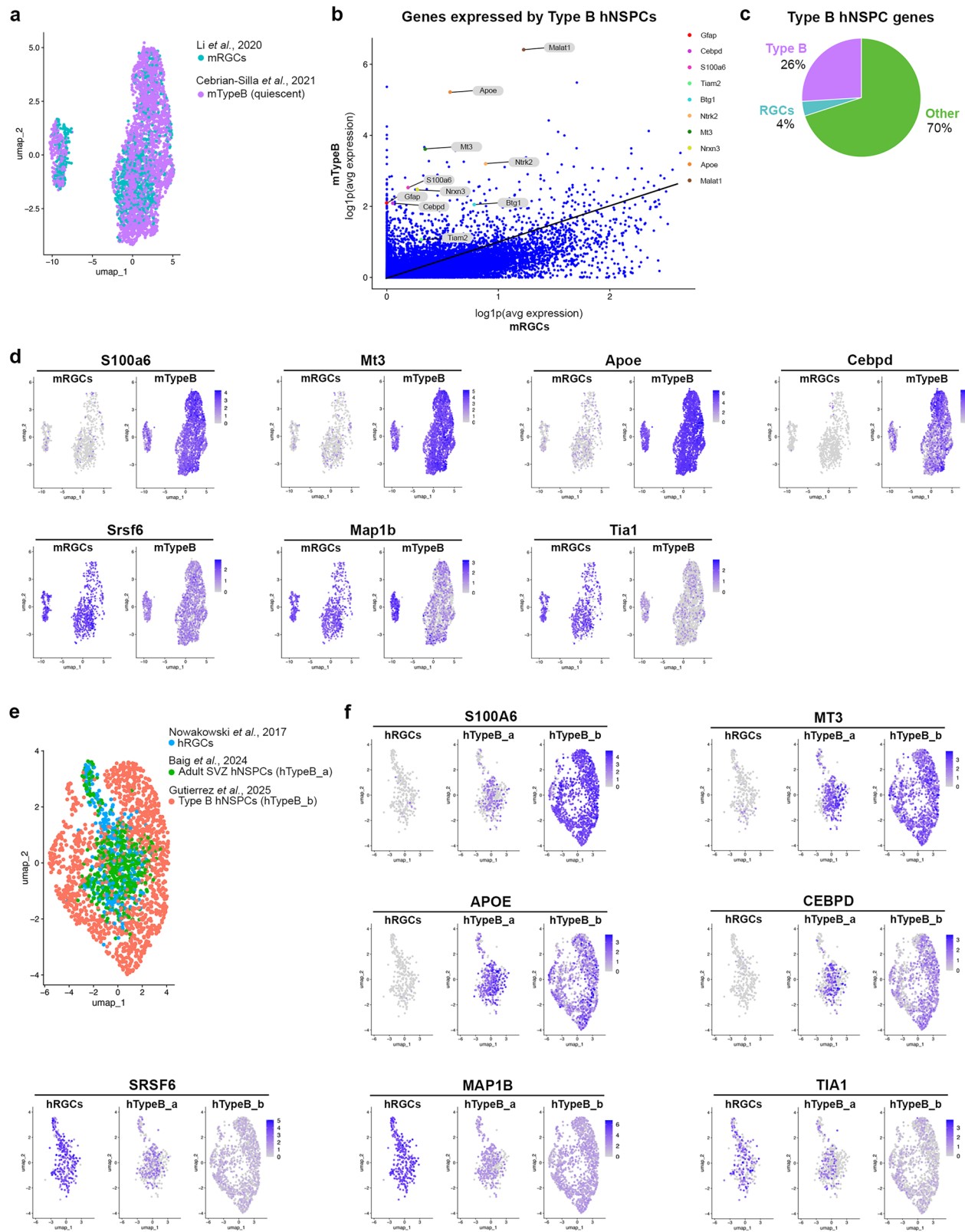

DLL4 stimulates Notch signaling in hNSPCs to increase the percentage of type B hNSPCs.

## Type B hNSPCs in the human SVZ co-express GFAP, SOX2, S100A6, LeX, and PROM1 and contact vasculature

In this study, we demonstrate hECs regulate hNSPCs via cell-cell contact in vitro. Although NSPC contact with vasculature in the rodent SVZ

niche is well-established[5,7,29], whether this cell-cell contact is observed in the human SVZ has not been closely studied. HNSPCs are found in the "astrocyte ribbon" identified by high GFAP expression adjacent to a gap lacking nuclei that separates hNSPCs from ependymal cells lining the lateral ventricle[37] (Fig. 8a). To assess localization of type B cell markers in vivo, human brain tissue across a range of ages (gestational week 36–15 years old) was stained for GFAP, SOX2, S100A6, LeX, and

**Fig. 4 | Type B hNSPCs express genes related to in vivo mouse and human adult type B NSPCs. a** UMAP plot of integrated mouse datasets including mouse radial glial cells (mRGCs) identified in Li et al.[63] and quiescent mouse type B cells (mTypeB) from Cebrian-Silla et al.[64]. **b** Plot of the log1p average expression of genes, represented by dots, in mouse radial glial cells (mRGCs) and mouse type B cells (mTypeB). The diagonal line indicates relative location of equally expressed genes, while genes above the line are higher in type B cells and below the line higher in RGCs. The labeled dots are genes highly expressed by our hNSPC Type B cluster. **c** Pie chart demonstrating the percentage of the top 50 type B hNSPC genes, based on average log2 fold change, highly expressed by mouse type B cells (purple), mouse RGCs (blue), and cells that are neither type B nor RGCs (green). **d** Plots demonstrate higher expression of S100a6, Mt3, Apoe, and Cebpd by mTypeB, while mRGCs more highly express Srsf6, Map1b, and Tia1. **e** UMAP of integrated datasets including human radial glial cells (hRGCs) identified in Nowakowski et al.[66], adult SVZ hNSPCs from Baig et al.[74] (hTypeB_a), and our type B hNSPCs (hTypeB_b). **f** Plots demonstrate S100A6, MT3, APOE, and CEBPD expression by hTypeB cells while hRGCs more highly express SRSF6, MAP1B, and TIA1, as observed in the mouse data. Source data are provided as a Source Data file.

PROM1 and imaged with confocal microscopy. The region containing the SVZ was identified (Supplementary Fig. 15) and higher magnification confocal imaging assessed marker localization in the tissue as well as co-expression in cells.

GFAP and SOX2 staining was prominent in the SVZ. GFAP and SOX2 co-expressing cells were identified in the SVZ at multiple stages (15-year-old, 6-year-old, and 2-year-old), showing that markers utilized in in vitro studies identify cells in the human SVZ (Fig. 8a, Supplementary Figs. 15a, 16). GFAP, S100A6, and LeX were all highly expressed in the SVZ of 15-, 6-, and 2-year-old tissue, while expression level and marker co-staining decreased in tissue further from the ventricle (Fig. 8b, 20x image, Supplementary Figs. 15b, 17, 18, 19). Higher magnification images revealed SVZ cells co-expressing these markers, which identify different sub-cellular compartments (GFAP is cytoskeletal, S100A6 is in the cytoplasm, and LeX is a carbohydrate on the membrane) (Fig. 8b, Supplementary Figs. 17, 18, 19). Astrocytes outside of the SVZ expressed GFAP but did not express S100A6 or LeX (Supplementary Fig. 18b). We stained human brain tissue for PROM1 since it was identified as a putative human type B cell marker in the in vitro model (Fig. 3, Supplementary Fig. 14) and GFAP and PROM1 co-expression increased in hNSPCs after hEC co-culture (Fig. 9a). PROM1 staining was evident in the SVZ colocalized with GFAP in 2-year-old (Fig. 9b, Supplementary Fig. 15c), 15-year-old, 6-year-old, and gestational week (GW) 36 human brain tissue (Supplementary Figs. 20, 21). Higher magnification images show co-localization of cytoskeletal GFAP and membrane PROM1 staining of SVZ cells (Fig. 9b, Supplementary Figs. 20, 21). Interestingly, several type B cell markers (SOX2, S100A6, LeX and PROM1) were also expressed by ependymal cells adjacent to the lateral ventricle. SVZ type B cells can be distinguished from ependymal cells by location and GFAP staining in combination with these other markers.

To determine whether type B cells in the SVZ niche are in contact with vessels, we stained human brain tissue for GFAP, SOX2, PROM1, and CD31. GFAP+ cells in the SVZ extended projections and contacted CD31+ vessels (Fig. 10a, Supplementary Figs. 15d and 22, Supplementary Movie 1). Staining with GFAP and SOX2 revealed co-expressing cells with GFAP+ processes in contact with CD31+ vessels (Fig. 10b, Supplementary Fig. 15e). Several GFAP+PROM1+ cells wrapped around CD31+ vessels (Fig. 10c, Supplementary Figs. 15f, 23, and 24, Supplementary Movies 2 and 3). These data indicate contact between hECs in vessels and cells expressing type B cell markers in the human SVZ.

These findings provide evidence that human type B cells in the SVZ co-express GFAP, SOX2, S100A6, LeX, and PROM1, and GFAP+SOX2+ and GFAP+PROM1+ type B cells contact vasculature in the human niche. Our scRNAseq data and human SVZ immunostaining identify markers of human type B cells, including S100A6 and LeX (Supplementary Table 1).

## Discussion

ECs are adjacent to NSPCs during brain development and maintain close contact with NSPCs in adult neural stem cell niches such as the SVZ[29,59]. Most research on EC regulation of NSPCs has focused on rodent systems and EC secreted factors. Human cell studies are lacking, and the role of EC contact on NSPCs has not been extensively investigated. Studying human cells is vital as there are species differences between rodent and human neural stem cell niches[36,39–41]. In this study, we found hEC contact and Notch signaling stimulates an increase in human type B cells identified via type B cell marker staining in vitro and in vivo and confirmed through scRNAseq (Fig. 10d).

Using scRNAseq, we determined that the hNSPCs expressing GFAP, SOX2, S100A6, and PROM1 after hEC co-culture were more similar to adult type B cells than to embryonic RGCs. We identified multiple markers of type B cells from mouse and human scRNAseq data (Fig. 4, Supplementary Data 4–6) including S100A6 previously identified as a marker of adult rodent NSPCs[48–50]. Additional type B cell markers include APOE, which regulates maintenance of adult rodent hippocampal NSPCs[58]. Genes more highly expressed by RGCs (both mouse and human), include MAP1B and SOX11, which are primarily expressed by neurons or neural progenitor cells during mouse nervous system development[83,84]. Expression decreases postnatally but remains in neurogenic regions in the adult mouse brain[83,84]. Another RGC marker, FOXG1, is involved in regulating NSPC proliferation during development and is upregulated in glioblastoma stem cells[85]. However, the role of the remaining identified markers in regulating NSPC function during development or in adulthood must be studied further.

We confirmed type B cell markers GFAP, SOX2, S100A6, LeX, and PROM1 by immunostaining of cells in vitro as well as human brain tissue in vivo in the SVZ region known to house adult hNSPCs[37]. Of note, studies in rodents indicate that adult type B cells are generated prenatally, and the hNSPCs in our in vitro studies were derived from prenatal stages[86]. Our scRNAseq analysis shows human type B cells can be further sub-clustered, suggesting distinct type B cell subsets. S100A6 is more highly expressed in one subset, and a similar S100A6 expressing subset was seen in the Baig et al. human type B cell scRNAseq data[74], where 33% of human type B cells express S100A6 (Supplementary Data 4). Our immunostaining data also suggests type B cell subsets; GFAP+SOX2+ type B hNSPCs comprised 64% of hNSPCs in co-culture, while GFAP+S100A6+ hNSPCs were 36% and GFAP+LeX+ hNSPCs were 16%. The heterogeneity and distinct subpopulations of human type B cells requires further study.

HEC co-culture almost doubled the percentage of type B hNSPCs (increased from 30% to 56%) at the expense of astrocyte progenitors (reduced two-fold, from 26% to 13%) and proliferating progenitors (reduced four-fold, from 17% to 4%). Functional assays confirmed the decrease in astrocyte progenitors since hNSPCs differentiated after co-culture formed fewer AQP4+ astrocytes. In rodent studies, EC contact reduced NSPC differentiation, which is consistent with the hEC contact-induced decrease in astrocyte progenitors[32,35]. HEC co-culture reduced hNSPC proliferating progenitors identified by scRNAseq, which was confirmed by hNSPC Ki67 staining and EdU incorporation after co-culture. Similarly, in rodent studies EC contact decreased mouse NSPC proliferation[32,35]. Collectively, the human proliferation and differentiation data fit a model whereby hEC contact maintains undifferentiated, quiescent type B hNSPCs.

HEC contact increased human type B cells in vitro and staining of the human SVZ revealed niche vasculature in contact with GFAP+ processes extending from GFAP+SOX2+ cells. Further, staining for PROM1 showed GFAP+PROM1+ type B hNSPCs contacting CD31+

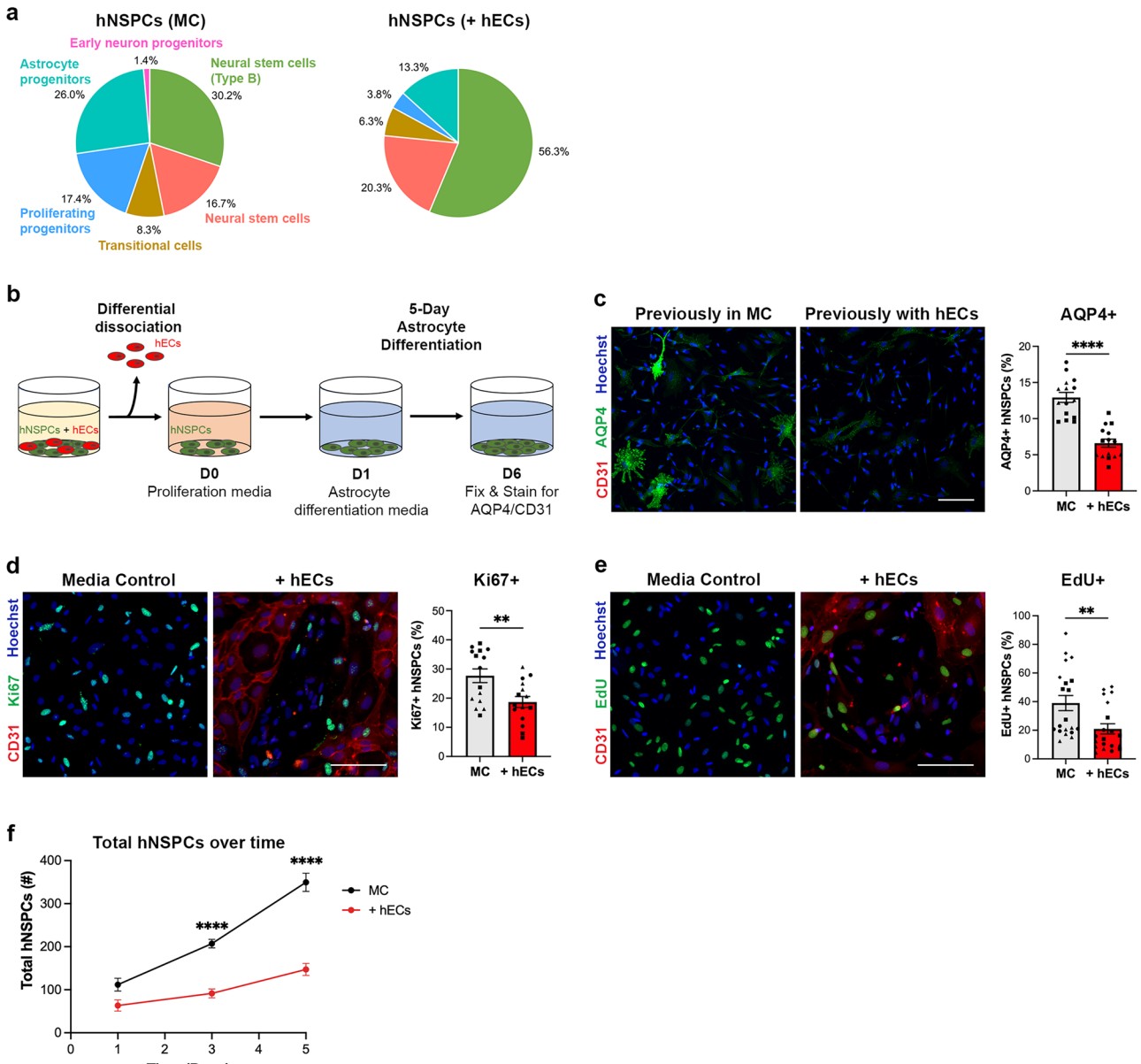

**Fig. 5 | hECs increase the percentage of type B hNSPCs while decreasing the percentage of astrocyte and proliferating progenitors. a** Pie chart demonstrating the percentage (calculated from scRNAseq data) of hNSPCs in each cluster in MC or with hECs. **b** Schematic of differential dissociation of hNSPCs after hEC co-culture to study astrocytic differentiation. Co-cultures (5 days) were trypsinized and since hECs were less adherent they dissociated first, leaving hNSPCs. Remaining hNSPCs were further dissociated, collected, plated on coverslips in proliferation media (day 0, D0), changed to astrocyte differentiation media and differentiated for 5 days. Coverslips were then fixed and immunostained for astrocyte marker aquaporin 4 (AQP4) and CD31 to ensure removal of hECs. **c** Differentiated SC27 hNSPCs previously in MC or hEC co-culture show the percentage of AQP4+ astrocytes significantly decreased in differentiated hNSPCs previously in co-culture (+hECs) compared to MC (****p < 0.0001). (d) SC27 hNSPCs

in MC and with hECs stained for the proliferation marker Ki67 and EC marker CD31 show the percentage of Ki67+ hNSPCs significantly decreased in +hECs compared to MC (**p = 0.0059). **e** SC27 hNSPCs in MC and with hECs after 24-h EdU incubation show the percentage of EdU+ hNSPCs significantly decreased in +hECs compared to MC (**p = 0.0076). **f** The total number of SC27 hNSPCs over time shows reduced hNSPCs in hEC co-culture with significant differences between MC and +hECs on day 3 (****p < 0.0001) and day 5 (****p < 0.0001). There was no significant difference in total number of hNSPCs between co-culture and MC at day 1 (p = 0.1821). Nuclei were stained with Hoechst. Scale bars, 100 µm, n = 3 (**c, d, f**) or 4 (**e**) independent biological replicates indicated by symbols, 5 areas quantified per sample, percentages are out of total hNSPCs, mean with SEM. Analysis used (**c–e**) unpaired two-tailed Student's t-test and (**f**) two-way ANOVA with Tukey post-hoc test for multiple comparisons. Source data are provided as a Source Data file.

vessels, indicating contact between hECs and hNSPCs in human brain. HECs regulate hNSPCs through contact-mediated Notch signaling and hEC contact up-regulates Notch downstream mediators HEY1 and HES4 in type B hNSPCs. Co-culture with hECs also stimulates Notch signaling in other hNSPC populations, and while HEY1 is highest in type B cells, HES4 is highest in astrocyte progenitors. There are non-redundant functions of HEY and HES transcription factors[87], so Notch signaling could have different roles in type B hNSPCs compared to

astrocyte progenitors. Inhibiting hNSPC Notch signaling with DAPT treatment reduced GFAP+SOX2+ type B hNSPCs in hEC co-culture. In addition, knocking down the expression of the Notch ligand DLL4 in hECs resulted in a decrease in type B hNSPCs in co-cultures. DLL4 is expressed exclusively by hECs and not hNSPCs, so DLL4 effects are only from hEC contact. DLL4 binds and activates NOTCH1 and NOTCH4[88], and hNSPCs express NOTCH1 but not NOTCH4. DLL4 was the only Notch ligand tested that increased GFAP, SOX2, and TNC

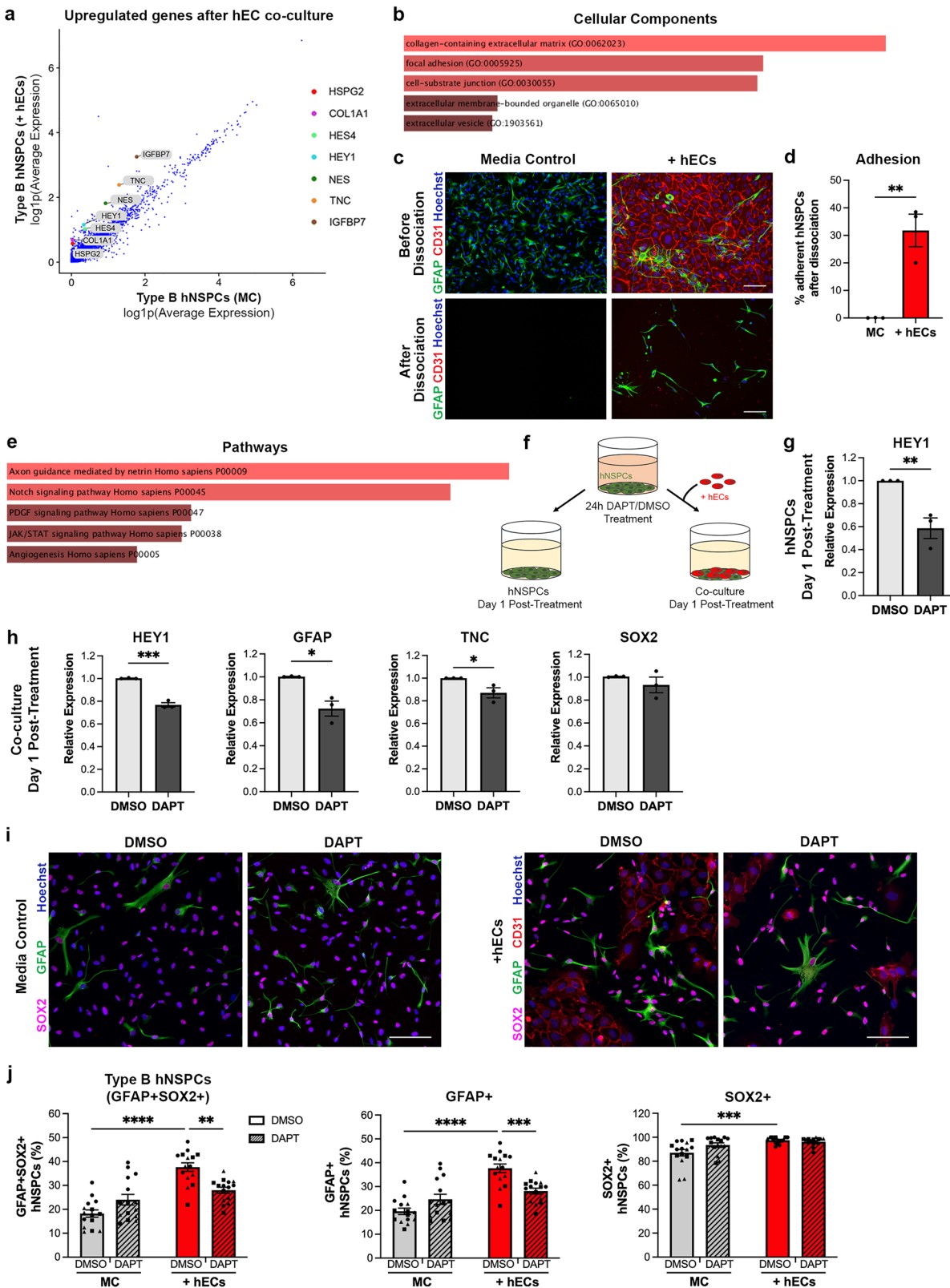

expression. Thus, hEC DLL4 can bind to hNSPC NOTCH1 to activate hNSPC Notch signaling and lead to an increase in type B hNSPCs.

Notch signaling effects on human type B cell phenotype were tied to changes in hNSPC GFAP expression and Notch has been previously tied to GFAP in rodent studies. Notch signaling increased GFAP expression in adult rat hippocampal-derived progenitors and many of these GFAP+ cells also expressed Nestin[89]. In addition, activation of the

Notch pathway in mice in vivo increased the number of GFAP+ cells in the SVZ[90] while inhibition of Notch in the adult mouse SVZ increased type B cell differentiation into transit-amplifying progenitors (type C cells) and neurons, resulting in type B cell depletion. This may relate to the hEC contact-mediated reduction in hNSPC astrocyte progenitors and suggest that Notch regulates human type B cell differentiation. Notch signaling regulates NSPC proliferation as well[91] and could play a

**Fig. 6 | Notch signaling is up-regulated in hNSPCs after hEC contact and inhibiting Notch signaling diminishes the effect of hECs on type B hNSPCs. a** Plot of the log1p average expression of genes in type B hNSPCs in MC compared to +hECs. Select genes up-regulated by type B hNSPCs in co-culture are labeled. **b** Cellular components gene enrichment analysis of up-regulated genes revealed collagen-containing extracellular matrix (adjusted p = 0.000240), focal adhesion (adjusted p = 0.000998), and cell-substrate junction (adjusted p = 0.000998). The longer and lighter the bar, the higher the significance for that term. **c** SC27 hNSPCs were grown in media control or hEC co-culture then dissociated, fixed and stained with GFAP and CD31. More GFAP+ hNSPCs remain in hEC co-culture compared to media control after dissociation. **d** Compared to MC, SC27 hNSPCs in hEC co-culture are more adherent to laminin (**p = 0.0058); values are percent remaining hNSPCs after dissociation. **e** Signaling pathways gene enrichment analysis of up-regulated genes highlighted axon guidance mediated by netrin (adjusted p = 0.0386) and Notch signaling (adjusted p = 0.0386). **f** Schematic of hNSPC pre-treatment with DAPT, a γ-secretase inhibitor, to block Notch signaling. HNSPCs treated with DAPT or DMSO for 24 h were plated in MC or + hECs for one day. **g** HNSPCs pre-treated with DAPT exhibited a significant reduction in HEY1 expression relative to DMSO

control (**p = 0.0099). **h** Compared to DMSO controls, DAPT pre-treated hNSPCs in co-culture had significantly reduced HEY1 (***p = 0.0003), GFAP (*p = 0.0132) and TNC (*p = 0.0431) expression. **i** HNSPCs pre-treated with DMSO or DAPT in MC or + hECs were immunostained for GFAP, SOX2, and CD31. **j** The percentage of GFAP +SOX2+ (****p < 0.0001), GFAP+ (****p < 0.0001), and SOX2+ (***p = 0.0004) hNSPCs significantly increased in hEC co-culture compared to MC in DMSO controls. DAPT treatment significantly reduced GFAP+SOX2+ (**p = 0.0013) and GFAP+ (***p = 0.0008) cells in hEC co-culture compared to DMSO co-culture control. Co-culture with DAPT treated cells did not significantly increase GFAP+SOX2+ (p = 0.3736), GFAP+ (p = 0.4579), and SOX2+ (p = 0.6635) cells compared to treated cells in MC. Relative expression (**g**, **h**) determined via qRT-PCR normalized to GAPDH. Nuclei were stained with Hoechst. Scale bars, 100 μm, n = 3 independent biological replicates indicated by symbols, 5 areas quantified per sample, percentages are out of total hNSPCs, mean with SEM. Analysis for (**d**, **g**, **h**) unpaired two-tailed Student's t-test, (**j**) two-way ANOVA with Tukey post-hoc test for multiple comparisons, (**b**, **e**) Fisher's exact test with Benjamini-Hochberg correction for multiple hypotheses testing conducted by Enrichr. Source data are provided as a Source Data file.

role in the hEC-induced reduction in hNSPC proliferation. HEC contact-mediated Notch signaling may help maintain a quiescent pool of hNSPCs in the human SVZ. However, whether Notch signaling after hEC contact reduces hNSPC differentiation or proliferation in vivo is not yet clear.

HNSPC adhesion was influenced in multiple ways by hEC contact. Firstly, hNSPCs formed clusters in hEC co-cultures that were not observed in monoculture, hEC CM, hEC transwell, or human lung fibroblast co-culture, suggesting a specific effect of hEC contact on hNSPC clustering. In rodent studies, EC contact induced NSPC migration towards each other and formation of NSPC clusters that were also not present in NSPC-EC transwells[32,35], suggesting similar NSPC clustering induced by EC contact across species. In addition to cell clustering, hNSPCs co-cultured with hECs increased adhesion to laminin-coated surfaces. HNSPCs express a number of α and β integrin heterodimers that mediate binding to particular extracellular matrix molecules[17,92], including heterodimers containing β1 integrins that bind laminin, fibronectin, or collagen, which are all produced by hECs. In addition, hEC co-culture increased type B hNSPC expression of genes affecting adhesion, including COL1A1 and TNC, a matrix component that also binds β1 integrin heterodimers. TNC expression appears to be regulated by Notch signaling as DLL4 increased hNSPC expression of TNC and inhibition of Notch signaling with DAPT reduced hNSPC TNC expression in co-cultures. Thus, the increased adhesion of hNSPCs after hEC co-culture is likely mediated by hNSPC β1 integrin binding to ECM laid down by both hECs and hNSPCs.

HEC secreted factors did not consistently affect hNSPCs in vitro, whereas hEC contact did. However, we cannot fully rule out short-range hEC secreted factors that may not be captured in CM or transwell studies. Further, hEC secreted factors may be present in vivo but dynamically regulated or not captured well in vitro; for example, early passage hBMEC CM increased GFAP+SOX2+ hNSPCs but that effect decreased rapidly with hBMEC passage. Distinct effects of CM from hECs and later passage (passage 3) hBMECs compared to early passage (passage 1) hBMECs could reflect differences in the tissue source of the cells, means by which they were isolated, or response to in vitro culture. The hBMEC data suggests a model whereby hEC contact is a stable inducer of the type B phenotype and secreted material could be more dynamic and labile. In the human in vivo niche, hNSPCs likely receive strong contact-mediated signals from hECs as well as soluble signals that vary in intensity.

ScRNAseq datasets will help clarify cell types in the human SVZ, particularly since there is often overlap in commonly used markers of cell phenotype. For example, human type B cells express SOX2, S100A6, LeX, and PROM1, but these markers also stained ependymal cells in the human SVZ and previous studies demonstrated

PROM1 staining of the human ependymal cell layer[93], suggesting some overlap between ependymal and type B cell markers in humans. Our type B hNSPC cluster identified via scRNAseq does not express ependymal cell markers, including FOXJ1 and PIFO[94], and coupled with recent adult human SVZ scRNAseq data, our scRNAseq dataset could help delineate type B cell-specific expression patterns[74]. In contrast with human cells, ependymal cells in the rodent SVZ are not positive for LeX[46] or S100A6[48]; however, scRNAseq analysis of rodent ependymal cells indicates S100A6 expression[64]. Further studies are needed to clarify expression differences between human type B cells and ependymal cells, particularly from cells in vivo.

A limitation of the current study is the relatively low numbers of hNSPCs in the co-culture scRNAseq data for technical reasons explained in Results. However, we obtained sequencing data on a sufficient number of hNSPCs to perform analysis and the scRNAseq results matched well with immunostaining data (percentage type B cells in co-culture scRNAseq: 56%, ICC: 64%). In addition, we verified scRNAseq data with subsequent independent assays. For example, scRNAseq data indicated decreased percentages of proliferating cells and astrocyte progenitors and we verified these data by immunostaining. Differential gene expression analysis from scRNAseq demonstrated an increase in type B hNSPC expression of Notch downstream mediators. We confirmed the role of Notch signaling by inhibiting Notch in hNSPCs via DAPT, reducing Notch ligand DLL4 in hECs by siRNA, and stimulating Notch signaling in hNSPCs with exogenous DLL4. The number of hNSPCs sequenced in future studies could be increased by analyzing 1:1 co-cultures of hNSPCs and hECs since we found this to be sufficient to increase type B hNSPCs (Supplementary Fig. 1g).

In summary, this study fully investigated the effect of hECs on hNSPC phenotype, proliferation, and differentiation. HEC contact causes an increase in human type B cells, and this study generated a human type B cell scRNAseq dataset. This dataset identified human type B cell markers and can help determine human type B cell clusters in scRNAseq data from adult human SVZ tissue. Analysis of human type B cell transcriptional profiles will be valuable for assessing signaling in the SVZ niche and further deciphering the role of hECs in regulating these interesting cells. Notch was determined as a key mechanism by which hEC contact stimulates human type B cells and future studies will continue to uncover the myriad ways Notch regulates hNSPCs. We found in previous studies hNSPCs stimulate hEC vessel formation, showing the complex bidirectional communication between these important cell types[17] (Fig. 10d). Understanding how hECs affect hNSPCs provides insight into the role of hECs in the adult human neural stem cell niche and will be critical for understanding human brain function.

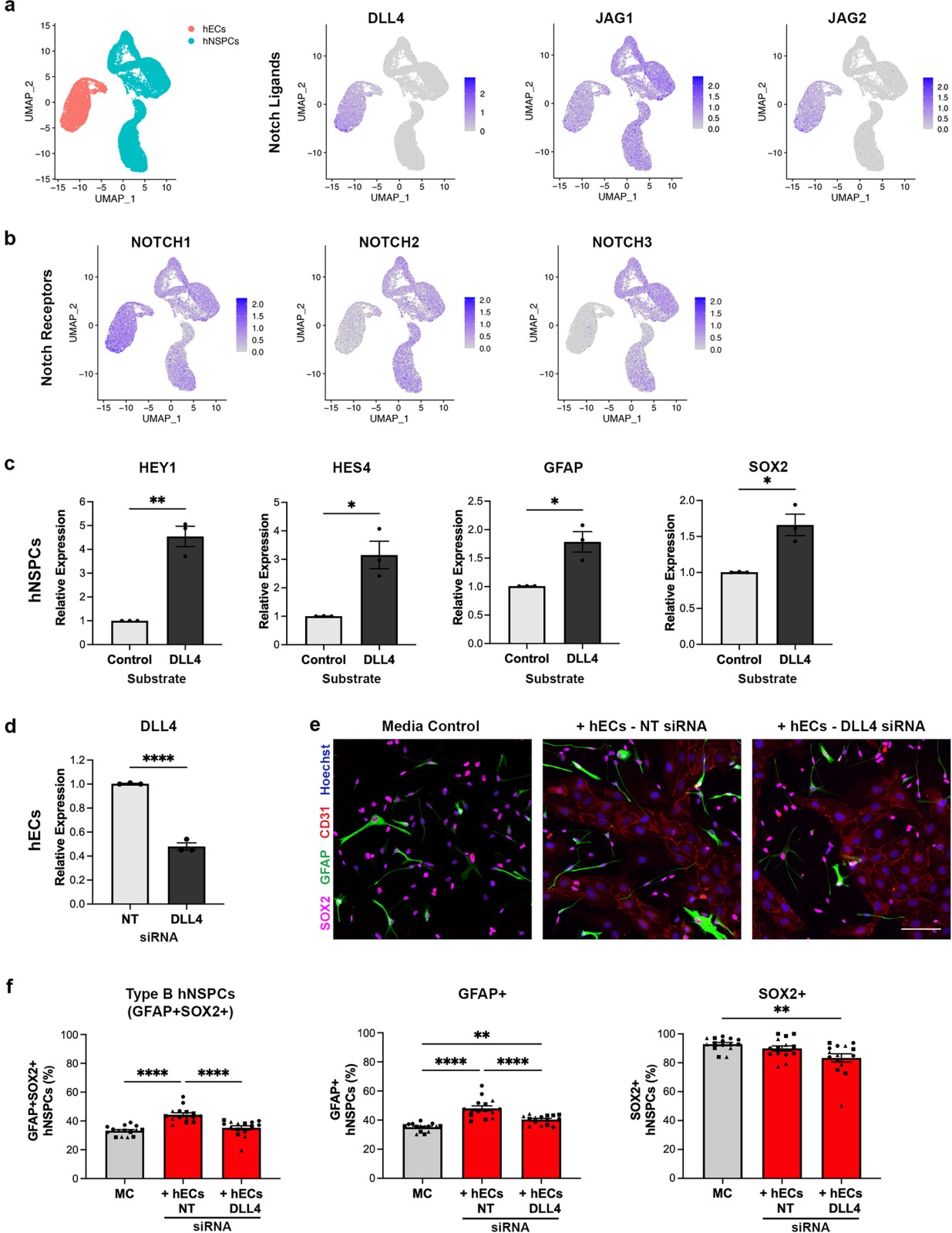

# Methods

## Cell culture

Human fetal brain-derived neural stem/progenitor cells were isolated from cadaveric brain cortices and designated as SC27 (male, 23 weeks gestation), SC30 (male, 25 weeks gestation), and SC23 (male, 23 weeks gestation)[95]. Use of human stem cells was approved by the University of California, Irvine Human Stem Cell Research Oversight (HSCRO) committee and earlier collection of human samples for cell isolation was approved for the National Human Neural Stem Cell Resource by the Children's Hospital of Orange County Institutional Review Board with informed consent and cells available for research purposes only. All tissues and cells were acquired in compliance with NIH and institutional guidelines.

**Fig. 7 | hEC DLL4 up-regulates type B cell markers and reducing DLL4 expression diminishes the effect of hECs on hNSPCs. a** UMAP demonstrating hEC and hNSPC clusters along with featureplots of Notch ligands DLL4, JAG1, and JAG2 expressed by hECs. **b** Featureplots showing the expression of Notch receptors NOTCH1, NOTCH2, and NOTCH3 by hNSPCs. **c** SC27 hNSPCs plated on DLL4 ligand increased the expression of HEY1 (**$p = 0.0011$), HES4 (*$p = 0.0114$), GFAP (*$p = 0.0123$), and SOX2 (*$p = 0.0115$) relative to control. **d** DLL4 expression was reduced in hECs treated with DLL4 siRNA compared to non-targeting (NT) control (****$p < 0.0001$). **e** SC27 hNSPCs in MC and co-culture with hECs treated with NT siRNA or DLL4 siRNA at day 1 after plating were stained for GFAP, SOX2, and CD31. **f** The percentage of GFAP + SOX2+ (****$p < 0.0001$) and GFAP+ (****$p < 0.0001$) hNSPCs significantly increased in NT siRNA-treated hEC co-cultures compared to MC. DLL4 siRNA-treated hECs significantly decreased GFAP + SOX2+

(****$p < 0.0001$) and GFAP+ (****$p < 0.0001$) hNSPCs compared to NT control. There was no significant difference between MC and DLL4-siRNA treated hEC co-cultures in terms of GFAP+SOX2+ type B hNSPCs (p = 0.4583). GFAP+ cells significantly increased in DLL4 siRNA-treated hEC co-cultures compared to MC (**p = 0.0079). SOX2+ cells significantly decreased in DLL4 siRNA-treated hEC co-cultures compared to MC (**p = 0.0058). Relative expression determined via qRT-PCR normalized to GAPDH (**c**) or RPLP0 (**d**). Nuclei were stained with Hoechst. Scale bars, 100 μm, $n = 3$ independent biological replicates indicated by symbols, 5 areas quantified per sample, percentages are out of total hNSPCs, mean with SEM. Analysis used (**c**, **d**) unpaired two-tailed Student's t-test and (**f**) one-way ANOVA with Tukey post-hoc test for multiple comparisons. Source data are provided as a Source Data file.

HNSPCs were cultured on 10 μg/mL human plasma fibronectin (EMD Millipore) coated plates as adherent cultures in proliferation media. HNSPC proliferation media is composed of DMEM/F-12 50/50 with L-glutamine and 15 mM HEPES (Corning), 20% (v/v) BIT-9500 (STEMCELL Technologies), 1% (v/v) antibiotic/antimycotic (penicillin/streptomycin/amphotericin; Thermo Fisher Scientific), supplemented with growth factors EGF (40 ng/mL; PeproTech), bFGF (40 ng/mL; PeproTech), and PDGF-AB (40 ng/mL; PeproTech). Cells were passaged 1:2 or 1:3 when 100% confluency was reached with Cell Dissociation Buffer (Thermo Fisher Scientific) or TrypLE Select (Life Technologies) for 8 min at 37 °C, neutralized with hNSPC proliferation media and washed with PBS. Cells were pelleted at $200 \times g$ for 5 min and resuspended in proliferation media. Media was changed 50% every day or 100% every other day.

Cord blood-derived human endothelial colony-forming cell-derived endothelial cells (hECs) were isolated as previously described[96], with informed consent and approval by UC Irvine's Institutional Review Board. Cells were cultured on 0.05% (w/v) porcine gelatin (Sigma-Aldrich) in VascuLife VEGF Endothelial Medium (Lifeline Cell Technology) as adherent cultures. HECs were passaged 1:2 or 1:3 when 90–95% confluency was reached using TrypLE Select (Life Technologies) for 1 min at 37 °C for dissociation neutralized by FBS. Cells were pelleted at $200 \times g$ for 5 min and resuspended in VascuLife media. Media was changed every 3 days. Human brain microvascular endothelial cells or hBMECs (ScienCell, male cells) were cultured the same as cord blood-derived hECs (female cells) except they were grown in the recommended Endothelial Cell Medium or ECM (ScienCell). Cord blood-derived hECs were used up to p8 and hBMECs were used up to p3.

Normal human lung fibroblasts (Lonza, female cells) were grown as adherent cultures in 10% FBS in DMEM (Gibco) in T25 flasks. Fibroblasts were dissociated with 0.25% Trypsin EDTA (Sciencell) which was neutralized by Trypsin Neutralizing Solution (ScienCell). Cells were centrifuged at $200 \times g$ for 5 min and resuspended in 10% FBS in DMEM. Cells were split 1:2 or 1:3 after 70–80% confluency was reached, and media was changed every other day. All cells were cultured in a humidified incubator at 37 °C and 5% $CO_2$.

## Co-culture plating
HNSPC (SC27, SC23) and hEC (cord-blood derived and hBMECs) co-cultures were plated at a 1:5 ratio (8000 hNSPCs:40,000 hECs). HNSPC (SC30s) and hEC co-cultures were plated at a 1:1 ratio (24,000 SC30s: 24,000 hECs). For hNSPC and NHLF co-culture, a 1:1 ratio was used for day 1 co-culture (48,000 SC27s:48,000 NHLFs) and a 4:1 ratio was used for day 5 co-cultures (24,000 SC27s:6000 NHLFs). All co-cultures were grown on 12 mm diameter glass coverslips pre-coated with 40 μg/mL poly-D-lysine (PDL, Sigma-Aldrich) in milliQ water for 5 min followed by 20 μg/mL natural mouse laminin (Thermo Fisher) in MEM at 37 °C overnight. HEC and NHLF co-cultures were grown in VascuLife media, while hBMEC co-cultures were grown in ECM. Co-cultures were grown

for 5 days unless otherwise specified, with 100% media change every other day.

For some experiments, hNSPCs and hECs were plated in 3D scaffolds as previously described[17]. Cells used for seeding were washed 3 times with Ca2+/Mg2+ free PBS and resuspended in MEM for counting and seeding. For a 50 μL gel, 250,000 hECs and 50,000 hNSPCs were used. Scaffold components were combined in an eppendorf tube with final concentrations of 5 mg/mL salmon fibrinogen (SeaRun Holdings), 1 mg/mL hyaluronic acid (Glycosil; BioTime), and 100 μg/mL laminin (Thermo Fisher Scientific). Cells were resuspended in MEM and added to the scaffold mixture for a total volume of 48 μL. To polymerize the gel, 2 μL of 100 U/mL salmon thrombin (SeaRun Holdings) was added to the cell and scaffold mixture. Scaffolds were allowed to polymerize for 30 min at room temperature before adding hEC media (VascuLife) dropwise around the gel to avoid stress to the gel. Media was changed 50% every day or 100% every other day using a pipette instead of aspirating to remove old media. After 5 days, gels were fixed in 4% paraformaldehyde for 20 min and washed 4 times for 15 min in PBS.

## Conditioned media
HEC conditioned media (CM) or hBMEC CM was collected after 3 days of cells in culture and centrifuged at $200 \times g$ for 5 min to remove cells. Supernatant was then centrifuged at $2300 \times g$ for 30 min to remove cell debris.

## Transwell co-cultures
12-well Transwell inserts (Corning) and 18 mm coverslips in the bottom compartment were coated with PDL/laminin before plating. 40,000 hECs or 50,000 hNSPCs (SC27s) were plated in the insert. 113,000 SC27s were plated on the 18 mm coverslip below the inserts. Co-cultures were plated at a 1 hNSPC to 5 hEC ratio (18,000 hNSPCs: 90,000 hECs). Co-cultures were grown for 5 days in VascuLife with media changes every other day.

## Neuronal differentiation
15,000 cells were plated on 12 mm coverslips pre-coated with PDL (40 μg/mL; VWR) diluted in milliQ water for 5 min followed by 20 μg/mL laminin (Thermo Fisher) in MEM (Thermo Fisher Scientific) overnight at 37 °C. HNSPCs were plated in proliferation media for 1 day, and then switched to neuronal differentiation media for 21 days with media changes every other day. Neuronal differentiation media is composed of DMEM/F-12 50/50 with L-glutamine and 15 mM HEPES (Corning), 20% (v/v) BIT-9500 (STEMCELL Technologies), 1x B27 Supplement (Thermo Fisher Scientific), 1× GlutaMAX Supplement (Life Technologies), 100 U/mL Penicillin-Streptomycin (Fisher Scientific), 20 ng/mL BDNF (PeproTech), 20 ng/mL GDNF (PeproTech), 0.5 μM dibutyryl cyclic AMP (Sigma-Aldrich). Media was changed every other day.

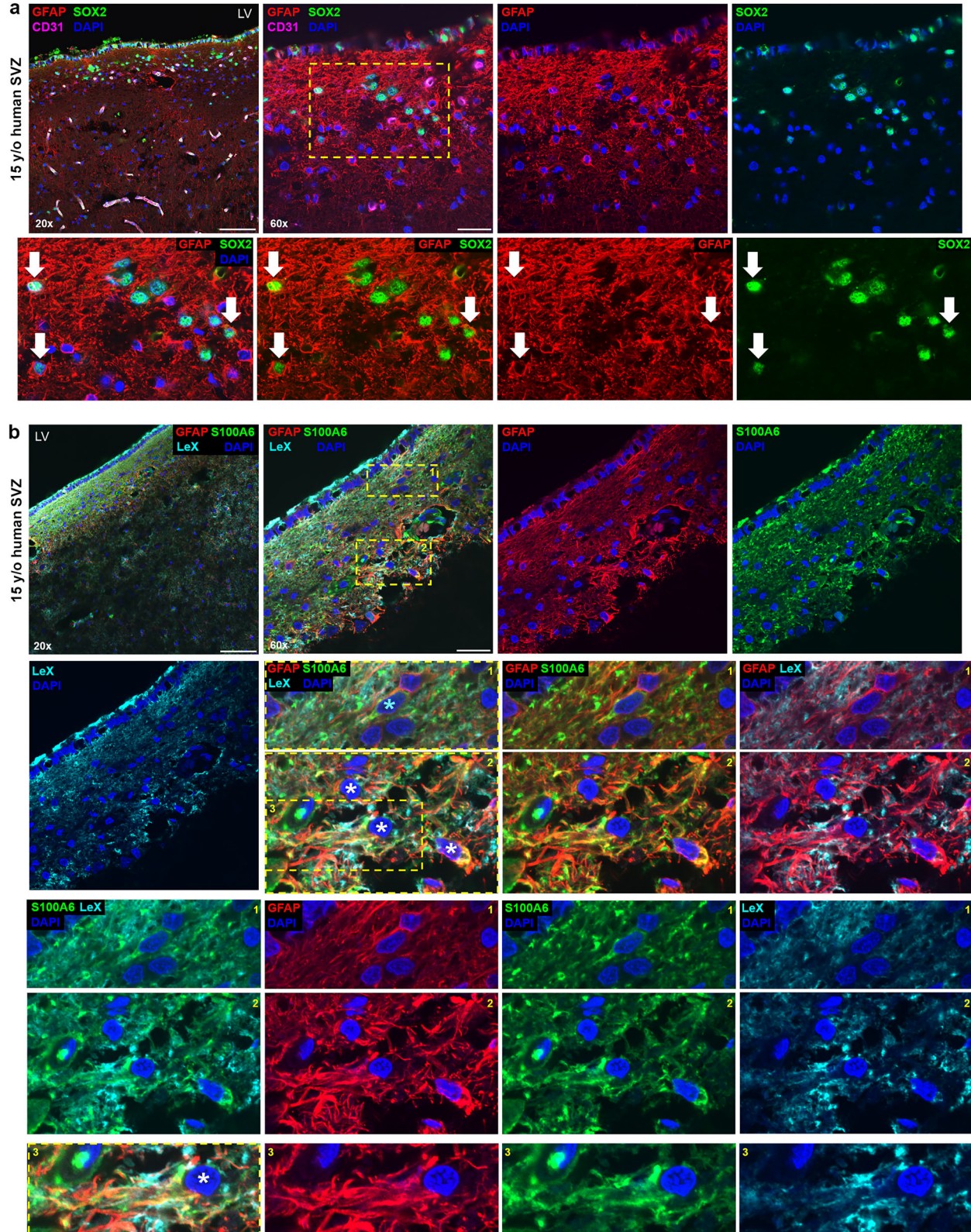

## Astrocyte differentiation

For hNSPC astrocyte differentiation, 50,000 hNSPCs were plated on 12 mm PDL/laminin coverslips and differentiated for 5 days in astrocyte differentiation media comprised of Neurobasal medium (ThermoFisher Scientific), 20% FBS, 1× B27 Supplement (Thermo Fisher Scientific), 1× GlutaMAX Supplement (Life Technologies), 100 U/mL Penicillin-Streptomycin (Fisher Scientific), 20 ng/mL BDNF

(PeproTech), 20 ng/mL GDNF (PeproTech), and 0.5 μM dibutyryl cyclic AMP (Sigma-Aldrich). Media was changed every other day.

## Co-culture differential dissociation for hNSPC astrocyte differentiation

Co-cultures were plated in one well of a 6-well plate coated with PDL/laminin at a 1:1 ratio (500,000 hNSPCs: 500,000 hECs) in VascuLife

**Fig. 8 | GFAP+ cells in the human SVZ co-express SOX2, S100A6, and LeX. a** 15-year-old human SVZ was stained for GFAP, SOX2, CD31, and DAPI and imaged with confocal microscopy ($n = 1$, biological repeats at other stages in Supplementary Fig. 16). Directly adjacent to the lateral ventricle (LV) is the ependymal cell layer visible as a line of adjacent nuclei, some expressing SOX2. Below the ependymal cell layer is a ribbon of GFAP+ and SOX2+ type B cells in the subventricular zone (SVZ) (20x image, scale bar 100 µm). Higher magnification 60x images show greater detail of GFAP+ and SOX2+ cells in the SVZ (scale bar 30 µm). Boxed area is further enlarged in lower panels to show individual cells, and examples of GFAP+, SOX2+ co-expressing cells are denoted by arrows. In multiple cases, basket-like GFAP+ structures encircle SOX2+ nuclei. **b** 15-year-old human SVZ was stained for GFAP, S100A6, LeX, and DAPI and imaged with confocal microscopy ($n = 1$, biological

repeats at other stages in Supplementary Figs. 18, 19). Lower magnification images (20×, scale bar 100 µm) show enrichment of signal in the SVZ. Higher magnification images (60×, scale bar 30 µm) show positively stained cells in the SVZ, and boxes denote spatially distinct regions enlarged further in lower panels. In the zoomed insets, the cell in box 1 labeled with a light blue asterisk co-expresses GFAP and S100A6, with GFAP adjacent to the nucleus and both markers extending into a process. Cells marked with white asterisks in box 2 co-express GFAP, S100A6, and LeX. These markers identify different sub-cellular compartments (GFAP cytoskeleton, S100A6 cytoplasm, and LeX membrane), and positive staining is present in non-nuclear regions of marked cells. Enlarged region (box 3, lower panels) shows a cell marked by asterisk co-expressing GFAP, S100A6, and LeX.

media. After 5 days of co-culture, hECs were removed by dissociating with TrypLE for 3 min at room temperature as they are less adherent than hNSPCs. Remaining adherent cells, mostly hNSPCs, were washed twice with PBS and dissociated with TrypLE for 8 min at room temperature followed by 1 min at 37 °C. Cells were centrifuged at $200 \times g$ for 5 min and the pellet was resuspended in proliferation media for plating on PDL/laminin-coated 12 mm coverslips (-1–2 coverslips for 1 well of 6-well co-culture). The day after plating, media was changed to astrocyte differentiation media and cells were differentiated for 5 days. For the media control, $1 \times 10^6$ hNSPCs were cultured in one well of a 6-well in VascuLife for 5 days, dissociated with TrypLE for 4 min at room temperature, plated on coverslips for differentiation at various densities (20,000–50,000 hNSPCs/12 mm coverslip) to determine a density to match that of the plated hNSPCs from the co-culture.

### SOX2 gene expression as hNSPCs differentiate
SC27 hNSPCs were plated on PDL/laminin coated coverslips and differentiated in Neurobasal, 2% B27 (Thermo Fisher Scientific), 1% GlutaMAX (Thermo Fisher Scientific), 1% pen/strep (penicillin/streptomycin; Thermo Fisher Scientific), with 20 ng/mL BDNF, 20 ng/mL GDNF, and 0.5 µM dibutyryl cyclic AMP (cAMP; Sigma-Aldrich). RNA was isolated from undifferentiated SC27s and those differentiated for 5, 10, or 15 days using a Bio-Rad RNA Isolation Kit. Extracted RNA was quantified using Nanodrop and cDNA synthesized using M-MLV reverse transcriptase (Promega). Quantitative real time-PCR (qRT-PCR) was performed using TaqMan primers (Thermo Fisher Scientific) on an Abi ViiA7 qPCR machine (Applied Biosystems) with 18S as housekeeping gene.

### Adhesion assay
Cell adhesion experiments were performed in a PDL/laminin coated 8-well chambered coverglass (Nunc Lab-Tek II, Thermo Fisher). HNSPCs or hNSPC-hEC co-cultures (1:1 ratio) were plated at -70 K cells/mL in 500 µL of VascuLife media and media change was performed every other day. On Day 5, cells were washed with 500 µL PBS then dissociated with TrypLE at room temperature for 2 min by pipetting up and down six times with a P1000 pipette. The enzyme was quenched with 500 µL conditioned media and 500 µL VascuLife media followed by a 2× PBS wash. Cells were allowed to recover overnight then were fixed for immunocytochemistry with GFAP and SOX2 to quantify adherent hNSPCs to determine percent adherent hNSPCs remaining after dissociation.

### Click-iT EdU
Click-iT EdU Alexa Fluor 488 Imaging Kit (Fisher Scientific) instructions were followed and cells on coverslips were incubated with 10 µM EdU for 24 h before fixing.

### Pre-DAPT treatment of hNSPCs for co-culture
500,000 hNSPCs were cultured in one well of a fibronectin-coated 12-well plate and treated with 10 µM DAPT (Calbiochem) or an equal volume of DMSO (Fisher Scientific) in proliferation media as a negative

control for 24 h before dissociation and plating alone (100,000 hNSPCs for media control) or with hECs at a 1:1 ratio (48,000 hNSPCs:48,000 hECs) on 12 mm PDL/laminin coated coverslips. Cells were fixed 1 day after plating.

### Notch ligand plating
Recombinant human Notch ligand Fc chimera proteins DLL4, JAG1, and JAG2 (R&D Systems) were plated on coverslips at a 20 µg/mL concentration along with laminin after PDL coating. Recombinant Human IgG1 Fc protein (R&D systems) at a 20 µg/mL concentration was used as a control. SC27 hNSPCs were cultured on ligand-coated plates for 5 days, dissociated, and RNA was isolated to conduct qRT-PCR as described below. For Fig. 7c, 5 µg/mL of DLL4 was used and hNSPCs were exposed to the ligand for 1, 3, and 5 days.

### DLL4 knockdown
250,000 hECs were plated in a 6-well plate and allowed to adhere for 1 day before treating with 1 µM Accell DLL4 siRNA SMARTpool (Horizon Discovery) for three days. RNA was collected for qRT-PCR confirmation of reduced DLL4 expression. After treatment, hECs were plated with hNSPCs at a 1:1 ratio (48,000 hECs: 48,000 hNSPCs) for 1 day, fixed, and stained for GFAP, SOX2, and CD31.

### Immunocytochemistry
Coverslips were fixed with 4% paraformaldehyde (4% paraformaldehyde, 5 mM MgCl$_2$, 10 mM EGTA, 4% sucrose in PBS) for 10 min, permeabilized with 0.3% TritonX-100 (Sigma-Aldrich) for 5 min, and blocked with 5% BSA (Jackson ImmunoResearch) for 1 h. Samples were incubated with primary antibodies diluted in 1% BSA at 4 °C overnight. For staining of cytoplasmic or nuclear markers in combination with membrane markers LeX or AQP4, a two-step primary antibody protocol was used where membrane markers were first stained without permeabilization. After primary antibody incubation overnight, stained cells were post-fixed with 4% paraformaldehyde for 5 min, permeabilized with 0.3% TritonX-100 for 5 min, and stained with cytoplasmic or nuclear markers of interest. Coverslips were then incubated in secondary antibodies diluted in 1% BSA (1:200) for 2 h in the dark at room temperature. Cells were counterstained with Hoechst 33342 nuclear dye (Life Technologies, H1399) at a 1:500 dilution in PBS for 5 min. Coverslips were mounted onto a slide using ProLong Diamond Antifade Mountant (Thermo Fisher Scientific).

For antibody staining of 3D scaffolds, cells in scaffolds were fixed with 4% paraformaldehyde for 20 min followed by $4 \times 15$ min PBS washes. Cells were permeabilized with 0.3% Triton X-100 (Sigma) in PBS for 15 min followed by $4 \times 15$ min PBS washes. Cells and scaffolds were then blocked using 10% donkey serum, 5% BSA, 0.1% Triton X-100 in PBS overnight at 4 °C on a rocker. Primary antibodies were diluted in 10% donkey serum, 5% BSA, 0.1% Triton X-100 in PBS and were incubated overnight at 4 °C on a rocker. Following primary antibody incubation, scaffolds underwent $4 \times 15$ min PBS washes. Secondary antibodies were diluted in 10% donkey serum, 5% BSA, 0.1% Triton X-100 in PBS and were incubated with cells for 2 h in the dark at room

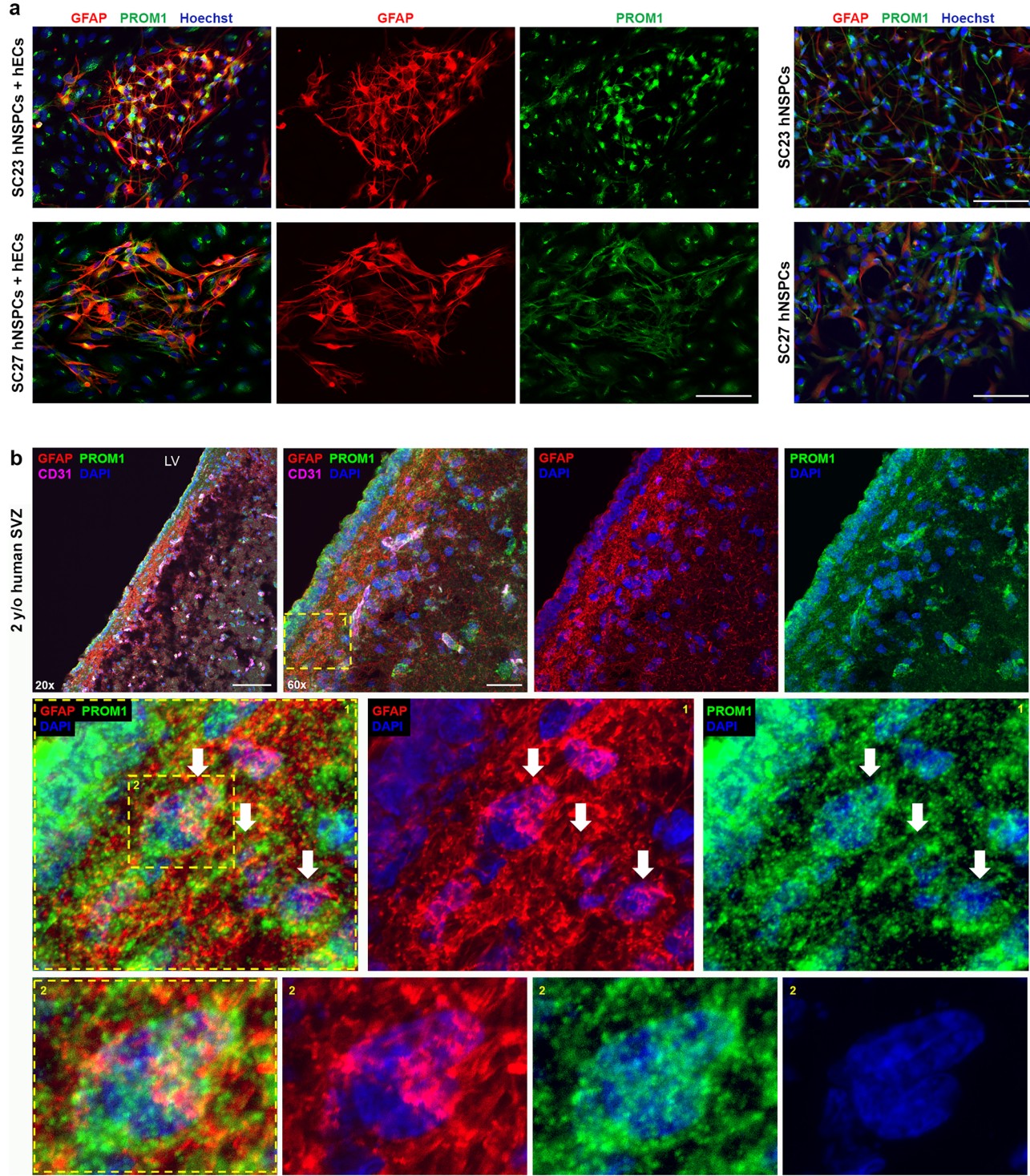

**Fig. 9 | Co-expression of GFAP and PROM1 in hNSPCs in vitro and in the human SVZ. a** SC23 (top row) and SC27 (bottom row) hNSPCs co-cultured for 5 days with hECs (left panels) or alone (right panels) and immunostained for GFAP and PROM1. GFAP +, PROM1+ cells are evident in co-cultures. Nuclei stained with Hoechst and scale bars 100 μm. **b** 2-year-old human SVZ was stained for GFAP, PROM1, CD31, and DAPI and imaged with confocal microscopy (*n* = 1, biological repeats at other stages in Supplementary Figs. 20, 21). Positive staining of the SVZ is evident in 20× images (scale bar, 100 μm). Higher magnification 60× images show the spatial distribution of GFAP and PROM1 and box denotes enlarged region shown in middle panels (scale bar 30 μm). Arrows in middle panels mark cells or clusters of cells with both GFAP and PROM1 (membrane) staining. The boxed cluster of cells is further enlarged in lower panels to show detail of GFAP and PROM1 co-staining.

temperature on a rocker. Secondary antibody incubation was followed by 4 ×15 min PBS washes. Cell nuclei were counterstained with 4 μg/mL Hoechst 3342 in PBS for 5 min, followed by 1×15 min PBS wash. Scaffolds were mounted with Vectashield (Vector Laboratories), a non-

hardening mountant. 8-well chambered coverglass with non-removable wells (Thermo Fisher Scientific) was used for easier imaging on an inverted microscope. Antibodies were as follows:

Anti-GFAP, rabbit IgG, DAKO Z033429-2,1:200

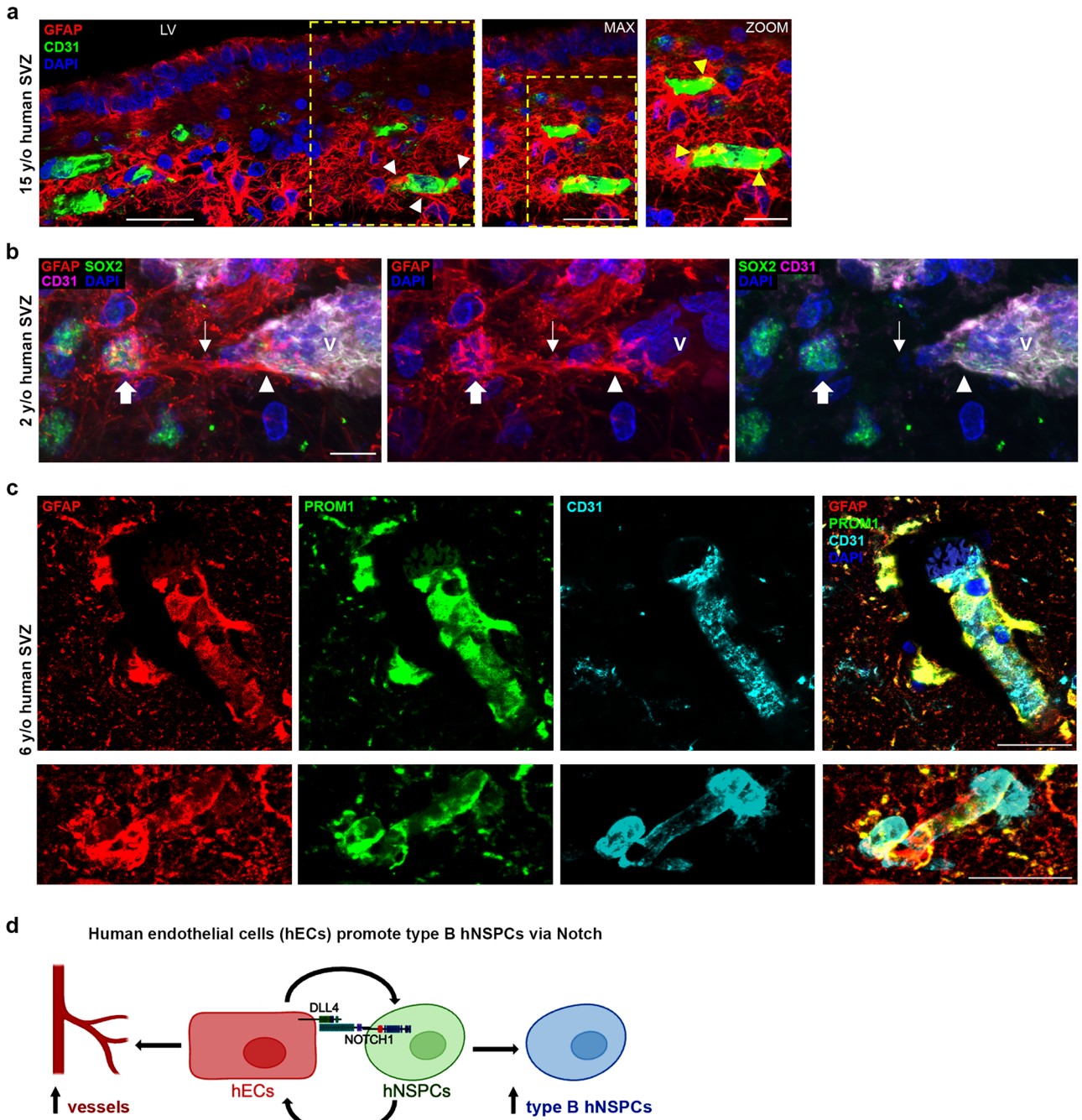

**Fig. 10 | GFAP+, SOX2+, and PROM1+ cells in the human SVZ contact vessels.**
**a** 15-year-old human SVZ was stained for GFAP, CD31, and DAPI and imaged with confocal microscopy (scale bar 50 µm) (*n* = 1, biological repeats at other stages in Supplementary Fig. 16). Arrowheads point to examples of GFAP+ processes contacting vasculature in the niche. Box denotes region for maximum projection of 3 confocal images (1 µm each, middle panel, scale bar 50 µm) to show overlap between GFAP+ processes and CD31+ vessels, which is shown in greater detail in zoomed panel (right, scale bar 25 µm) and denoted by yellow arrowheads. **b** 2-year-old human SVZ was stained for GFAP, SOX2, CD31, and DAPI and imaged with confocal microscopy (*n* = 1, biological repeats at other stages in Supplementary Fig. 16). SOX2+ nucleus surrounded by GFAP is denoted by the fat arrow, while GFAP+ process extending from the cell is marked by the thin arrow. The GFAP+

process contacts the CD31+ vessel (V) along the region denoted by the arrowhead. Scale bar 10 µm. **c** 6-year-old human SVZ stained for GFAP, PROM1, CD31, and DAPI and imaged with confocal microscopy (*n* = 1, biological repeats at other stages in Supplementary Figs. 20, 21). Processes from cells co-expressing GFAP and PROM1 extend toward and wrap around CD31+ vessels. Bottom right panel represents maximum projection of 9 confocal images (1 µm each). Scale bars, 30 um. **d** Schematic summarizing hEC-hNSPC reciprocal communication. HEC contact increases the percentage of hNSPCs with a type B cell phenotype via Notch signaling involving the Notch ligand DLL4 on hECs and Notch receptor on hNSPCs. As shown previously, hNSPCs increase hEC vessel formation, creating a positive feedback interaction between these critical cell types[17]. Created with BioRender.com.

Anti-GFAP, goat IgG, Abcamab53554,1:200
Anti-GFAP, mouse IgG, Sigma-Aldrich G3893, 1:200
Anti-SOX2, goat IgG, R&D SystemsAF2018, 1:200
Anti-CD31, mouse IgG, DAKO M0823, 1:200

Anti-LeX, mouse IgM,BD Biosciences 550382, 1:50
Anti-S100A6, rabbit IgG, Abcam ab181975, 1:100
Anti-AQP4, rabbit IgG, Sigma-Aldrich AB3594, 1:100
Anti-MAP2, mouse IgG, Sigma-Aldrich M9942, 1:200

Anti-DCX, goat IgG, Santa Cruz Biotechnology, SC-8066, 1:200
Anti-Ki67, rabbit IgG, Leica Biosystems, NCL-Ki67p, 1:500
Anti-Cleaved Caspase 3, rabbit IgG, Cell Signaling Technologies, 9661S, 1:200
AlexaFluor 488 donkey anti-rabbit IgG, Jackson ImmunoResearch, 711-545-152, 1:200
AlexaFluor 594 donkey anti-mouse IgG, Jackson ImmunoResearch, 715-585-151, 1:200
AlexaFluor 633 donkey anti-goat IgG, Thermo Scientific A-21082, 1:200
AlexaFluor 647 donkey anti-mouse IgM, Jackson ImmunoResearch, 715-605-020, 1:200
AlexaFluor 594 donkey anti-goat IgG, Jackson ImmunoResearch, 705-545-147, 1:200
AlexaFluor 488 donkey anti-mouse IgG, Jackson ImmunoResearch, 715-585-151, 1:200

## Microscopy and quantitation of stained cells

Cells were imaged using a Nikon Eclipse Ti-E fluorescence microscope at 20× magnification and acquired using NIS Elements AR 4.51 image capturing and analysis software. Keyence BZ-X810 inverted widefield fluorescent microscope was also used for imaging cells at 20×. Some cells and scaffolds were also imaged using confocal microscopy as described below for human SVZ tissue.

Five random regions (each 750 µm × 563 µm) on the coverslips were imaged for quantitation. To ensure sufficient hNSPCs were captured in co-cultures for counting, Hoechst and CD31 fluorescence was used to identify regions with non-CD31 expressing cells, which are hNSPCs. This allowed for the unbiased identification of regions in co-cultures high in hNSPC numbers while not visualizing GFAP, SOX2, or other hNSPC-specific markers. Intensity thresholding was utilized to count medium and high expressing cells using cell counter on Image J. For S100A6, we counted all S100A6+ cells including low-expressing cells because scRNAseq data showed lower S100A6 expression by type B hNSPCs compared to non-type B hNSPCs. In some co-culture experiments, hECs and hNSPCs were distinguished by CD31 staining (hECs CD31-positive and hNSPCs CD31-negative).

## Human SVZ tissue

Autopsy, post-mortem (less than 48 h) gestational week 36, 2-year-old, 6-year-old, and 15-year-old human anterior horn SVZ tissue used as neuropathological controls was obtained at University of California, San Francisco (UCSF) following institutional guidelines with previous patient consent and strict observance of the legal and institutional ethical regulations set by the UCSF Committee on Human Research. Protocols were approved by the Human Gamete, Embryo and Stem Cell Research Committee (Institutional Review Board GESCR# 10-02693) at UCSF. Tissue was fixed in 4% PFA, then gradually switched to a sucrose gradient from 10% to 30% and embedded in OCT before freezing at −80 °C. Coronal sections 30 µm thick were cut on the cryostat.

## Immunohistochemistry

Slides were equilibrated to room temperature for 2.5 h, baked at 60 °C for 30 min, and washed with TNT buffer (PBS, 0.05% Triton X-100) to rehydrate tissue for 10 min. Sections were post-fixed with 4% PFA for 20 min and rinsed with TNT buffer, followed by water. Sections were incubated in 1% $H_2O_2$ for 30 min to block endogenous peroxidase then rinsed with water. Slides were placed in antigen retrieval solution (10 mM sodium citrate buffer, pH 6.0) for 10 min at 90–95 °C and then cooled for 30 min. Slides were then rinsed with water, quenched with 1% $H_2O_2$ for 1.5 h, and washed 3 times with TNT for 10 min. Tissue was blocked with TNB blocking buffer (0.1 M Tris-HCl, pH 7.5, 0.15 M NaCl, 0.5% TSA Blocking Reagent, Perkin Elmer) for 1 h. Sections were incubated with primary antibodies overnight at 4 °C. Primary antibodies

were sequentially incubated to avoid cross-reaction. After 3 TNT washes for 10 min, tissue was incubated with biotinylated secondary antibodies for 2.5 h at room temperature and washed with TNT 3 times for 10 min. All antibodies were diluted in TNB solution. Next, sections were incubated in Streptavidin-HRP (1:200 in TNB) for 30 min, followed by 3 TNT washes. Tyramide signal amplification (PerkinElmer) was used by incubating sections in tyramide-conjugated fluorophores (fluorescein 1:50, Cy3 1:100, Cy5 1:100) for 5 min and DAPI was used to stain nuclei. After several PBS rinses, sections were dehydrated, mounted with Fluoromount-G® Slide Mounting Medium, Southern Biotech/VWR catalog # 100241-874, and coverslipped. Antibodies were as follows:

Anti-GFAP, chicken IgY, Abcam, ab4674, 1:500
Anti-SOX2, goat IgG, R&D Systems, AF2018, 1:200
Anti-S100A6, rabbit IgG, Abcam, ab181975, 1:200
Anti-LeX, mouse IgM, BD Biosciences, 550382, 1:200
Anti-CD31, mouse IgG, DAKO, M0823, 1:200
Anti-PROM1, rabbit IgG, Invitrogen, PA5-38014, 1:200
Biotinylated donkey anti-rabbit IgG, Jackson ImmunoResearch, 711-065-152, 1:200
Biotinylated donkey anti-chicken IgY, Jackson ImmunoResearch, 703-065-155, 1:200
Biotinylated donkey anti-mouse IgM, Jackson ImmunoResearch, 715-065-140, 1:200
Biotinylated donkey anti-mouse IgG, Jackson ImmunoResearch, 715-065-150, 1:200
Biotinylated donkey anti-goat IgG, Jackson ImmunoResearch, 705-065-147, 1:200

## Confocal imaging of human SVZ

Human brain tissue including the SVZ region was imaged on an Olympus FV3000 confocal laser scanning microscope using 10× and 20× UPLSAPO objectives or with 40× or 60× UPLSAPO oil objectives. Images were taken with excitation wavelengths 405 nm, 488 nm, 561 nm, and 640 nm at 4096 × 4096 scan size using the Galvano scanner at a 2.0 µs/pixel scan speed in a one-way line sequential mode. Z-stack images at 20×, 40×, and 60× were obtained with step size of ~1 µm. Image J was utilized to view z-stacks and create z-stack videos. Stitched images to establish location of zoomed images were taken on a Keyence VZ-X810 inverted widefield microscope using the 4× objective and DAPI filter wheel.

## Single-cell RNA sequencing analysis

**Sample preparation.** Cells for single-cell RNA sequencing were prepared on the same day using the same cell sources for each condition to reduce batch effects. Cells were dissociated, resuspended in 0.4% BSA and a cell strainer was used to ensure single cells.

**Library preparation, sequencing, and mapping.** 10x Genomics v3.1 chemistry was utilized to create cDNA libraries. The libraries were then sequenced using Illumina NovaSeq 6000 with a target sequencing depth of 50,000 reads per cell: hNSPCs media control (11,903 cells with 47,374 mean reads per cell), hNSPCs in hEC CM (14,462 cells with 37,469 mean reads per cell) and hNSPC-hEC co-culture (9559 cells with 67,135 mean reads per cell). FastQC was used to initially check the quality of sequencing data. Sequence alignment to the human reference genome GRCh38 2020-A was performed using 10x Genomics pipeline Cell Ranger count v6.0.1.

**Analysis with Seurat.** Seurat[97] package version 4.0.4 with R version 4.1.0 was used for quality control, log normalization, and clustering of data. For quality control, cells with less than 700 genes and more than 7500 genes detected were excluded for analysis as well as cells with more than 12% mitochondrial DNA. After quality control, we analyzed 10,780 hNSPCs in media control, 13,871 in hEC conditioned media, and

co-cultures with 7375 hECs and 158 hNSPCs. Log normalization, scaling of data, and linear dimensional reduction were performed. For cell clustering, datasets were merged due to low batch effect and dimensionality was reduced using UMAP. HNSPC and hEC clustering used 6 principal components (PCs) and a resolution of 0.1, as shown in Fig. 3a. PCs were chosen based on elbow plots and heatmaps demonstrating differences in gene expression. Find All Markers was utilized to find genes expressed in each cluster. Subset analysis was conducted to identify additional hNSPC or type B hNSPC clusters and to calculate the percentage of cells in each cluster. Subset analysis of hNSPCs only (excluding hECs) as shown in Fig. 3c used 8 PCs and resolution of 0.2, and hNSPC clusters were identified using markers shown in Fig. 3d. Subset analysis of type B hNSPCs as shown in Supplementary Fig. 10 used 8 PCs and resolution of 0.1. Differential gene expression of type B hNSPCs in co-culture compared to media control was conducted using Find Markers. The type B hNSPC cluster in co-culture contained sufficient cells for differential gene expression analysis since the minimum is 75 to obtain better accuracy and low false positive rates and this cluster included 89 cells[98]. We did not conduct differential gene expression analysis on clusters with less than 75 cells. Cell-cycle scoring of scRNAseq hNSPC clusters was conducted using Seurat to identify cells in G1, G2/M, and S phases as demonstrated in Supplementary Fig. 7c. For plots demonstrating gene co-expression in Supplementary Fig. 10c, a color or blend threshold of 0.5 was utilized with a minimum and maximum cutoff value for each feature set at quantiles 10 and 90, respectively.

**Mouse dataset integration.** Mouse radial glial cell (RGC) clusters from the Li et al. study[63] included two dorsal (mRGC1 and mRGC2) and one ventral (mRGC3) RGC clusters from E15.5 mice. Mouse type B cell dataset was from the Cebrian-Silla study[64] containing quiescent dorsal and ventral type B cell clusters from P29 and P35 mice. The quiescent mouse type B cell dataset was utilized as human type B cells are likely largely quiescent as the human SVZ has fewer proliferating progenitors compared to the rodent SVZ and there is no evidence of neuroblasts in the human SVZ[36,40,41]. Before merging datasets, gene symbols from each dataset were changed to current gene symbols from the Mouse Genome Informatics (MGI) database (code provided in GitHub). For analysis of the Li et al. study using Seurat version 5.1.0 and R version 4.4.2., cells with more than 750 genes but less than 3750 genes and less than 5% mitochondrial DNA were kept for analysis. Normalization, scaling of data, and linear dimensional reduction were performed. A subset of mRGC1, mRGC2, and mRGC3 were obtained for integration. For the Cebrian-Silla et al. study, cells with more than 600 genes but less than 7500 genes and less than 5% mitochondrial DNA were kept for analysis. Normalization, scaling of data, linear dimensional reduction, and UMAP were performed using 10 PCs and a resolution of 1.2. Quiescent type B cell clusters were identified using genes described in the original study[64] and a subset of quiescent type B cells was saved for integration. Mouse RGC and type B cell subsets were merged and integration was performed using CCA integration. UMAP was utilized for visualization with 30 dimensions and resolution of 0.6 as demonstrated in Fig. 4a. Differential gene expression analysis was conducted using Find Markers. For the pie chart in Fig. 4c, the top 50 type B hNSPC genes (based on average log2 fold change) were considered either type B or RGC genes if the log2 fold change was 1 or −1 respectively, at least 25% of cells express the gene and that percentage is higher than the other cell type.

**Human dataset integration.** Human radial glial cell (hRGC) clusters were obtained from Nowakowski et al. study[66] including early radial glia, ventricular radial glia, outer radial glia, and truncated radial glia from 5.85 to 22 post-conception weeks. Human adult (ages 38–72 years old) SVZ NSC-like cells from Baig et al.[74] were identified from raw data using Seurat package version 5.1.0 and R version 4.4.2. Before merging

datasets, gene symbols from each dataset were changed to approved gene symbols from the HUGO Gene Nomenclature Committee (HGNC) database (code provided in GitHub). Analysis was conducted on cells from the Nowakowski study with greater than 200 and less than 6000 genes, as well as less than 8% mitochondrial gene expression. Log normalization, scaling of data, and linear dimensional reduction were performed. A subset consisting of tRG, vRG, oRG, and RG-early clusters identified in the original study[66] was saved for integration. Analysis was conducted on cells from the Baig et al. study with greater than 200 and less than 3500 genes, as well as less than 8% mitochondrial gene expression, as stated in the paper. Log normalization, scaling of data, and linear dimensional reduction were performed. For clustering, 35 PCs with a resolution of 0.6 were chosen and UMAP was utilized for dimensionality reduction. Find All Markers identified genes expressed in each cluster and the NSC-like cluster was identified using markers noted by Baig et al.[74]. The NSC-like cluster was isolated and merged with Nowakowski et al. hRGCs and our type B hNSPCs found in media control. In the merged dataset, cells with more than 200 and less than 6000 genes with less than 8% mitochondrial gene expression were used for analysis. Normalization, scaling of data, linear dimensional reduction, and CCA integration were performed. Clustering was conducted using 20 PCs with a resolution of 2. Find Markers was used to compare gene expression between hRGCs and NSC-like cells. A small number of mitochondrial genes were omitted from Supplementary Data 4, due to inconsistency in mitochondrial gene names between datasets.

### Gene list enrichment analysis

Web-based Enrichr using Gene Ontology (GO) cellular components and Panther metabolic and cell signaling pathway databases[75–77] was conducted of the top 50 up-regulated genes (adjusted $p$ value) of type B hNSPCs in co-culture compared to media control.

### Quantitative real time-PCR

RNA was isolated using Aurum Total RNA Mini Kit (Bio-Rad Laboratories). Complementary DNA was synthesized using M-MLV reverse transcriptase (1:25), 1× reverse-transcriptase buffer (Promega), 0.3 µM random hexamer primer (Thermo Fisher Scientific), 0.5 mM dNTPs (Fisher Scientific), and 0.08 U/µL RNase inhibitor (Fisher Scientific) with dilutions carried out using DEPC-H₂0. S100 Thermal Cycler (Bio-Rad) was used for cDNA synthesis. For a 10 µL reaction, 5 µL of PowerUp SYBR Green Master Mix (Thermo Fisher Scientific), 10 ng cDNA, 0.5 µL of 10 µM forward and reverse primers (IDT), and RNAase-free water was used. Primers were customized using NCBI's PRIMER-BLAST unless cited. QRT-PCR was conducted following SYBRgreen instructions on the QuantStudio 6 or 7 RT-PCR systems using an annealing temperature of 60 °C. Negative reverse transcriptase and/or negative cDNA controls were used for quality control. Data was analyzed by the comparative cycle ($C_t$) method[99] using the housekeeping genes GAPDH or RPLP0 for normalization. Primers were as follows

Gene: HEY1, Forward Primer (5′→ 3′)CCT TCC CCT TCT CTT TCG GC, Reverse Primer (5′ → 3′) AAA AGC TCC GAT CTC CGT CC, length (bp) 125

Gene: HES4, Forward Primer (5′→ 3′)ATC CTG GAG ATG ACC GTG AG, Reverse Primer (5′ → 3′) CGG TAC TTG CCC AGA ACG, length (bp) 88

Gene: GFAP, Forward Primer (5′→ 3′)AGG ATG GAG AGG AGA CGC AT, Reverse Primer (5′ → 3′) GAG CTC CAT CAT CTC TGC CC, length (bp) 231

Gene: SOX2, Forward Primer (5′→ 3′)AAC CAG CGC ATG GAC AGT TA, Reverse Primer (5′ → 3′) GAC TTG ACC ACC GAA CCC AT, length (bp) 278

Gene: TNC, Forward Primer (5′→ 3′)GTT AAC GCC CTG ACT GTG GT, Reverse Primer (5′ → 3′) CCA CAA TGG CAG ATC CTT CT, length (bp) 157

Gene: S100A6, Forward Primer (5′→ 3′)CTC CCT ACC GCT CCA AGC, Reverse Primer (5′ → 3′) CTG GAA GTT CAC CTC CTG GTC, length (bp) 244

Gene: PROM1, Forward Primer (5′→ 3′)TCA ATG ACC CTC TGT GCT TG, Reverse Primer (5′ → 3′) AAG ACG CTG AGT TAC ATT GTC G, length (bp) 296

Gene: DLL4, Forward Primer (5′→ 3′)CGG GTA CCT TCT CGC TCA TC, Reverse Primer (5′ → 3′) CAC ATA GTG GCC GAA GTG GT, length (bp) 263

Gene: GAPDH, Forward Primer (5′→ 3′)TGC ACC ACC AAC TGC TTA, Reverse Primer (5′ → 3′) GGA TGC AGG GAT GAT GTT C, length (bp) 177

Gene: RPLP0[100]

Forward Primer (5′→ 3′) GCT TCG TGT TCA CCA AGG AGG A, Reverse Primer (5′ → 3′) GTC CTA GAC CAG TGT TCT GAG C, length (bp) 135

## Graphs and statistical analysis

GraphPad Prism version 9.3.1 was utilized to make graphs and conduct statistical analysis. Bar graphs and line graphs show mean and standard error of the mean. Comparison of two samples utilized two-tailed unpaired Student's t-tests. Datasets containing more than two samples were analyzed by one-way ANOVA with Tukey's post hoc correction for multiple samples. Datasets with two variables were analyzed by two-way ANOVA with Tukey post-hoc test for multiple comparisons. For scRNAseq differential gene expression analysis using Seurat, Wilcoxon Rank Sum test with Bonferroni correction was utilized for calculation of adjusted $p$ values. Enrichr analysis uses Fisher's exact test with Benjamini-Hochberg correction for calculation of adjusted $p$ values.

## Reporting summary

Further information on research design is available in the Nature Portfolio Reporting Summary linked to this article.

## Data availability

The scRNAseq data generated in this study have been deposited in the NCBI Gene Expression Omnibus (GEO; http://www.ncbi.nlm.nih.gov/geo) database under accession code GSE255624. Additional data generated in this study are provided in the Supplementary Information/Source Data file. Source data are provided with this paper.

## Code availability

The codes used in this study have been deposited in GitHub: https://github.com/bgutie1/Nature-Communications-2025.git.

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

## Acknowledgements

The authors gratefully acknowledge Dr. Christopher C.W. Hughes for providing the hECs used in this study as well as Dr. Phillip Schwartz for the hNSPCs. We thank the UCI Genomics High Throughput Facility for processing and sequencing cells for the scRNAseq study and Dr. John W. Lau for providing codes to change gene symbols to those approved in MGI and HGNC databases. Appreciation to Dr. Kai Kessenbrock and Dr. Edwin S. Monuki along with their lab members for input on the scRNAseq analysis. This work was supported in part by the National Institutes of Health and National Science Foundation under award numbers NIH F31 NS118845 (B.G.), NIH T32 NS082174 (B.G.), UCI Medical Scientist Training Program and NIH T32 GM008620 (B.G.), NIH R01 NS119829 (L.A.F.), NSF IOS-2019400 (L.A.F.), NIH R61/R33 HL154307 (L.A.F.), NIH P01 NS083513 (M.F.P), NIH DP2 NS122550-01 (M.F.P), as well as the Roberta and Oscar Gregory Endowment in Stroke and Brain Research (M.F.P) and the Chan Zuckerberg Initiative (M.F.P). The content is solely the responsibility of the authors and does not necessarily represent the official views of the funding agencies, including the National Institutes of Health.

## Author contributions

Conceptualization, B.G., L.A.F.; Methodology, B.G., L.A.F.; Investigation, B.G., C.R., T.C.L., O.P.-A., H.L.; Resources, M.F.P, L.A.F.; Writing—Original Draft, B.G.; Writing—Review and Editing, L.A.F.; Funding Acquisition, B.G., L.A.F.; Supervision, L.A.F.

## Competing interests

The authors declare no competing interests.
