## [Transparent Peer Review file · Nature Communications]

Human endothelial cells promote a human neural stem cell type B phenotype via Notch signaling

Corresponding Author: Dr Lisa Flanagan

Version 0:

Reviewer comments:

Reviewer #1

(Remarks to the Author)

This study investigated the interaction between hNSPCs and hECs using an in vitro co-culture model. The results showed that contact with ECs promote human neural stem cell type B phenotype (GFAP+Sox2+). Using scRNA-seq, they identified various cell types in hNSPCs when they are in contact with ECs, and found that type B phenotype is enriched in EC co-culture condition. Further studies revealed the gene expression in the putative type B cells, and further focus on the role of Notch signaling pathway as the main driver of type B cell types. This study is significant as it provides important human neural stem cell data that are different from rodent studies.

While the RNA-seq data is compelling, the choice to use SOX2 and GFAP together to stain for hNSPCs reduces the impact of the ICC results. Specifically, the authors continuously rely on the assumption that 'most astrocytes are GFAP+SOX2-' (Line 130), when in fact proliferating and reactive astrocytes highly express SOX2, and some studies have even found that astrocytes constitutively express SOX2. This would indicate that GFAP+SOX2+ positive stains may not be a reliable combination to selectively identify type B NSPCs, making it difficult to have complete confidence in all NSPC percentage data that was obtained using this stain. Additionally, the Notch pathway is known to regulate proliferation of reactive astrocytes, in which TNC and GFAP are also involved. Notch inhibition could be decreasing the number of reactive astrocytes, also decreasing the expression of TNC, GFAP, and SOX2. There is not a proposed control to show that these cells are not astrocytes and the Notch inhibition effect observed is selective to NSPCs.

The conclusion of this study is that EC contact via Notch signaling confer the hNPCs type B phenotype. Because scRNA-seq data does not provide spatial information, and some of the staining in the manuscript are difficult to see the spatial localization of these cells and their relation to blood vessels, it will be important to show the spatial localization of these cells (GFAP+Sox2+, GFAP+Lex+, GFAP+S100A6+), in area close to blood vessel vs. those far away from blood vessels. Because all these cells and ECs are evenly dispersed in the hydrogel, it may be difficult to quantify the percentage of these with regards to the distance to blood vessels. One potential solution is using EC spheroid (which will generate long sprout in hydrogel) instead of the dispersed ECs. Spatial distribution of type B cells in relationship to the vessel would be easily seen. Other potential solution is to use organ-on-chip model of vascular system in which vessels are better organized or can be designed with spatial patterns.

For the Notch signaling, it will be important to provide direct evidence that EC and hNSPCs interact directly by co-staining the Notch ligand in ECs and receptor in hNSPCs, and demonstrate the spatial distribution of these ligand-receptor pairs.

Figure 5f. showed that in co-culture condition, the total number of hNSPCs is less than control, suggesting less proliferation when co-cultured with ECs. However, this could be simply because there are more cells in this condition (vs. control condition) which make it crowded for cells to grow. Additional control experiments (e.g. fibroblasts co-culture) are needed to show that this effect is not simply because of crowd but specific to endothelial effect.

Figure 8a, b, the staining of GFAP, Lex, S100A6 is essentially everywhere, this type of staining is not very useful to tell their relationship with blood vessels.

(Remarks to the Author)

Gutierrez et al. aim to verify in a human setting observation from rodents in which contact with ECs regulate hNSPC homeostasis. The authors suggest that ECs promote a type B phenotype through a Notch-mediated mechanism based on immunofluorescence staining, analysis of single-cell RNA sequencing, and functional experiments. While these studies were performed in human-based in vitro 2D cultures, the authors claim the presence of hNSPCs type B cells in postnatal human brain tissue.

Major

-The authors report an increase in GFAP+SOX2 cells upon hEC co-culture in Figure 1. It is however unclear what the authors mean with % GFAP+SOX2+ cells in Figure 1. Is this amount normalized over the number of SOX2+ cells per condition (1b, 1c, 1d, 1e)? This is needed to accurately assess the extent of increase of GFAP+SOX2+ cells as the amount of SOX2+ cells increases as well (1b).

I furthermore wonder about the classification of marker positivity across figure 1. For example, in panel 1c the main difference between media control and +hECs seems to be the intensity of GFAP expression rather than individual cell positivity. The representative image shown does not indicate 20% GFAP+SOX2+ in MC. More representative lower magnification images for all conditions would be useful to be able to evaluate the robustness of this phenotype. Panels in Supplementary Figure 1d should be overlaid with SOX2.

The initial amount and percentage of GFAP+SOX2+ cells should likewise be shown at day 0 (Supplementary 1d and 1e) to understand the starting point. Could the authors comment why there is already a high percentage of GFAP+SOX2+ cells in the media control condition at day 1? Do these data not rather indicate a maintenance of GFAP+SOX2+ cells (Supplementary 1d) rather than increasing GFAP+SOX2+ cells induced by hECs?

-Figure 2: More representative and lower magnification images as well as single channel and overlaid panels of the same areas should be shown to evaluate the presence of LeX, GFAP and double positive cells across conditions. Also here, I wonder about the classification of marker positivity, given that it is difficult to see how the quantified stark increase in the GFAP+LeX+ cells matches the representative images. The same holds true for the S100A6 staining. These data (including quantifications) should be repeated with SC30 hNSPCs to increase robustness of the observations.

-As a general note to most quantifications in the paper: what is meant with wording like "5 areas analyzed". How large are these areas? How were they chosen? Can the authors color the dots belonging to the same biological experiments in the quantitative graphs so that it is easier to understand the data robustness?

-The authors report mixed evidence upon conditioned media experiments (Supplementary Figure 2-3). The authors should show overlaid panels of GFAP and SOX2 in Supplementary Figure 2, at high and low magnification. This data is currently derived from n=1 and needs to be repeated across lines and independent experiments and include statistical tests. As per before, normalization of GFAP+SOX2+ over total SOX2+ is also needed. Figure 1c and 1d would likewise strongly benefit from including the conditioned media data from Supplementary Figure 3 to the main figures, to put the data variation across the observed effects in the different experimental settings in a more transparent manner.

Could the authors attempt to biologically discuss these differences? Finally, the authors should show consistent effects of the direct hEC co-culture over later passages, since the authors argue that CM are not consistent due to the passaging effect, but the reverse is not shown.

-Figure 3-4: The single-cell RNA sequencing data is heavily biased to cells derived from the hNSPC mono-culture and hNSPCs with hEC conditioned media, while very few hNSPC cells are retrieved in the hNSPCs-hEC co-culture which is arguably the most relevant setting (as mentioned by the authors in the legend of Supplementary Fig. 6a). Increasing the numbers of the hNSPCs in the hEC co-culture conditions is however needed to reach more robust conclusions, especially when attempting to infer cell type distributions across datasets.

SOX2 and LeX (CD15) should be included in the dot plot in fig. 1d to further understand type B classification.

Based on the identification now made of the human type B cells, could the authors now identify this population in available single-cell RNA sequencing datasets of human (fetal) brain? This would greatly strengthen the in vitro observations.

-Figure 5: Based on which marker staining did the authors identify and quantify (total) hNSPCs in 5c-f? Figure 5c seems to show weakly positive AQP4 cells in the previous EC condition. Does this rather indicate impaired differentiation?

-Figure 6-7: The authors postulate an importance of Notch signaling in the type B phenotype upon hEC co-culture and experiments with DAPT corroborate this. To strengthen these observations, experiments also with SC30 hNSPCs would increase data robustness. Furthermore, performing the same DAPT experiments in a conditioned medium set-up would further strengthen the need for direct contact, rather than other effects of DAPT. There seems a general variation in GFAP+SOX2+ across experiments performed. Why is there a marginal increase from 35 to 40% in GFAP+SOX2 hNSPCs in 7f upon hEC with NT siRNA, while this was increased earlier from 20 to 60% in 1b? This questions the robustness of the siRNA experiments and effects observed, since the hEC co-culture experiment seems to not increase significantly GFAP+SOX2+ already by itself.

Since conditioned medium experiments also influenced type B phenotype with varied observations, the authors could further investigate the importance of secretion by e.g. blocking general cell secretion with small molecules. It is important that the authors place the importance of Notch in the right context keeping in mind their observations across conditioned medium and co-culture experiments.

-Figure 8: The evidence for this cell type in vivo as it stands is rather weak. The authors should also stain for SOX2 in the tissue. Why do the authors further suddenly switch to PROM1 for the tissue stainings? Do the in vitro cells also show PROM1 positivity in the GFAP+SOX2+ cells? Do additional markers identified from Fig. 3-4 show similar overlap? How do the authors explain the abundant number of PROM1+ cells that are not into contact with CD31+ cells? Finally, could the authors stain for active Notch signals, e.g. DLL4, in relation to the proximity of the CD31+ cells in the tissue to strengthen their putative mechanism?

Reviewer #3

(Remarks to the Author)

General comments: In this manuscript, with the reasoning that differences exist between the postnatal human SVZ might differ from the mouse SVZ, the authors analyze the effect of hECs on hNSPCs using an invitro system. However, later in the study they describe that human type B NSCs resemble mouse TypeB NSCs. They describe hECs regulate quiescence of hNSCs (type B) via Notch signaling, which requires direct cell-cell contact. They also indicate that they have identified new human type B NSCs new markers, which they then try to validate them via immunostainings in human postnatal samples.

Specific comments:

- The co-culture system is in a 2D setting, which does not really reflect the in vivo situation. The findings using this 2D system should be validated using a 3D one, where vessel-like structures and NSCs association to vessels is present.
- Figure 1a-e: the text for CD31 inside the images is not visible, please change it to another location. In addition higher magnification and higher resolution images should be shown to show real EC-NSC contacts.
- Figure 2: other studies have shown that ECs secrete factors (i.e. BDNF, IGF1, etc) that induce proliferation and differentiation of NSCs. From the data presented in figure 5, it seems that in the settings used by the authors, hECs conditioned medium does not lead to increase NSC proliferation compared to medium control. Why is this? Is this system then not reflecting similar conditions are used before in other studies? It would to show that their settings can reproduce previously described mechanisms.
- Figure 2 b and e: the ECs should also be shown via staining for a EC marker. How are the contacts of type B NSCs with ECs?
- Figure 2e: it is very difficult to see the colocalization of S100A6 with GFAP. It is recommended that images for single channels are also shown.
- Line 227-231: According to this explanation, there are two clusters of endothelial cells: proliferative and non proliferative endothelia cells. It would be important and relevant for this research to study which kind of interactions are mainly represented between NSC-proliferative EC and which kind of NSC-non-proliferative EC interactions. Which ones lead to typeB NSCs?
- In lines 271-283, the GFAP is first stated as type B cell specific, then it is stated that GFAP is "well-known" to be expressed by both RGCs and type B cells, then at the end again described as type B specific.
- In line 272, NTRK2 is cited as type B specific, as shown in Fig4b. On Fig4b it seems that NTRK2 is expressed by both cell types present. Same goes, arguably, for BTG1 and TIAM2.
- Figure 4: this figure with the comparison between human type B NSPCs and mouse type B NSPCs seems to be rather isolated from the study. Would it not make sense to have it earlier to show the differences between both vertebrates? Considering the overall lower expression profile of mRGCs compared to mTypeBs (Fig4b), Fig4d could use some control genes that are known specific mRGC markers.
- Fig4f does not show any noticeable difference between the 2 subtypes depicted for the genes shown.
- Fig4g does not demonstrate, as it is written in line 297 that S100A6 demonstrates the "greatest co-expression with GFAP", as no other co-expressions are shown. Also, the colour scale makes it extremely hard to determine the true co-expression of these genes. What units are shown in the co-expression legend?
- Figure 5a: please indicate from where are those pie charts calculated? Are they from the scRNAseq data?
- An important point is the graph shown in Fig5e, which claims a significantly higher EdU+ cell population in the media control compared to the co-cultures, but the graph shows very similar data populations with what seem 4 clear outliers in the control group.
- Fig6b and Fig6e: no axis and units shown for the bar graphs.
- Fig6c seems to be column-normalised for the average expression, so the true difference between the 2 conditions can not be estimated.

- Fig6d shows the "# adherent hNSPCs after dissociation (relative)". Since the values are fractions, then these are not the number of cells as the "number sign" (#) would suggest. What is it relative to?

- Figure 6j: the text for CD31 inside the images is not visible, please change it to another location. In addition higher magnification and higher resolution images should be shown to show real EC-NSC contacts.

- Fig6k : x axis legends for hEC are not clear, which one is DMSO and which one DAPT?

- Figure 8: Here the authors use human tissue to show that type B cells contact blood vessels and that they express LeX, S100A6 and PROM1. The quality of the images and the resolution of them does not allow to state their claims.

Version 1:

Reviewer comments:

Reviewer #1

(Remarks to the Author)

Reviewer's comments have been addressed.

Reviewer #2

(Remarks to the Author)

The authors have appropriately addressed most of my concerns. While in the revised version the authors have not increased the numbers of hNSPC cells from the EC-coculture in the single-cell RNA sequencing, which I do consider a shortcoming, I would advise the authors to transparently show this in the main body of the paper, regarding the amount of cells retrieved per condition (i.e. including S11a-b to the main figure 3) with a brief remark on this in the main text. Of note, the paper has considerably increased in length and amount of figures, which may need to be reconsidered, e.g. Fig.9 and 10 can be merged, and parts of Figure 4 can go into the supplement.

Response Type B paper reviews

We appreciate the reviewer's comments and recognition of the importance of the work. Our revisions to the manuscript include multiple new main figures, additions to figures, and supplementary figures, which bolster our findings and strengthen the robustness of our study. We have addressed reviewer comments as described below, with reviewer comments in italicized text.

Reviewer 1

(1) Concerns that some astrocytes express SOX2, complicating the immunocytochemistry analysis using SOX2.

While the RNA-seq data is compelling, the choice to use SOX2 and GFAP together to stain for hNSPCs reduces the impact of the ICC results. Specifically, the authors continuously rely on the assumption that 'most astrocytes are GFAP+SOX2-' (Line 130), when in fact proliferating and reactive astrocytes highly express SOX2, and some studies have even found that astrocytes constitutively express SOX2. This would indicate that GFAP+SOX2+ positive stains may not be a reliable combination to selectively identify type B NSPCs, making it difficult to have complete confidence in all NSPC percentage data that was obtained using this stain. Additionally, the Notch pathway is known to regulate proliferation of reactive astrocytes, in which TNC and GFAP are also involved. Notch inhibition could be decreasing the number of reactive astrocytes, also decreasing the expression of TNC, GFAP, and SOX2. There is not a proposed control to show that these cells are not astrocytes and the Notch inhibition effect observed is selective to NSPCs.

Response: We delved into the literature and SOX2 is expressed by some reactive and proliferating astrocytes in mice. In human scRNAseq data, astrocytes express SOX2, but at a much lower level than radial glial cells in the developing brain. We revised the wording in the Results section to remove the phrase "most astrocytes are GFAP+SOX2-" to reduce confusion. The scRNAseq data in our study confirmed the identity of cells co-expressing GFAP and SOX2 as type B hNSPCs using analysis of overall gene expression, which is beneficial for cell identification since markers can often overlap across different cell types. No astrocytes were identified in our cultures by the scRNAseq analysis. Furthermore, we utilized multiple type B cell markers in immunocytochemistry in addition to SOX2, including LeX and S100A6, which showed the same trends as observed with SOX2 (Figs. 1 and 2). We also utilized another type B cell marker, PROM1, and found that two different sets of hNSPCs (SC23 and SC27) co-cultured with hECs co-expressed GFAP and PROM1; this data is now included in new Figure 9. Overall, we used multiple markers (GFAP, SOX2, LeX, S100A6, and PROM1) for ICC *in vitro* and staining of human tissue *in vivo* as well as scRNAseq analysis that measures a cell's full expression profile to identify type B cells in the study.

(2) Important to provide spatial information and evidence of contact between hECs and hNSPCs.

The conclusion of this study is that EC contact via Notch signaling confer the hNPCs type B phenotype. Because scRNA-seq data does not provide spatial information, and some of the staining in the manuscript are difficult to see the spatial localization of these cells and their relation to blood vessels, it will be important to show the spatial localization of these cells (GFAP+Sox2+, GFAP+Lex+, GFAP+S100A6+), in area close to blood vessel vs. those far away from blood vessels. Because all these cells and ECs are evenly dispersed in the hydrogel, it may be difficult to quantify the percentage of these with regards to the distance to blood

vessels. One potential solution is using EC spheroid (which will generate long sprout in hydrogel) instead of the dispersed ECs. Spatial distribution of type B cells in relationship to the vessel would be easily seen. Other potential solution is to use organ-on-chip model of vascular system in which vessels are better organized or can be designed with spatial patterns.

Response: Reviewers requested additional evidence for contact and the spatial relationship between hNSPCs and hECs. We performed new experiments using cell surface markers for hNSPCs and hECs since this would allow us to observe cell perimeters and assess cell contact. We used LeX (CD15), which recognizes a cell surface glyco-epitope, for hNSPCs and CD31, which is platelet endothelial cell adhesion molecule (PECAM1) that is expressed on endothelial cell plasma membranes for hECs. Confocal imaging of the stained cells showed hNSPCs spread over hECs and revealed sites of contact between the two cell types. These new images are now included in Figure 2.

As a second approach, we co-cultured hNSPCs and hECs in three dimensional (3D) scaffolds mimicking brain mechanical and extracellular matrix properties. We found increased type B marker expression (GFAP and LeX) for hNSPCs co-cultured with hECs in 3D, and confocal imaging showed contact between hNSPCs and hECs. These new data are now included in Supplementary Figure 4.

Quantifying the precise spatial location of hNSPCs relative to blood vessels in human brain tissue is not possible because the sections are only 30 μm thick (need to be this thickness for antibody staining) and thus do not fully encompass vessels extending in 3D through the tissue. This makes analysis of hNSPCs relative to blood vessels challenging, because while we might mark an hNSPC as not in contact with a vessel based on that section, we will be unable to ascertain whether that cell is in fact contacting vessels outside of the plane of that section. In animal models, issues such as these have been addressed with tissue clearing protocols. However, human SVZ tissue is too scarce to allow that type of approach. As described in #5 below, we include new high resolution images of the human SVZ showing contact between type B cells and vasculature *in vivo*.

(3) Provide evidence that hECs and hNSPCs interact directly by co-staining the Notch ligand in hECs and receptor in hNSPCs.

For the Notch signaling, it will be important to provide direct evidence that EC and hNSPCs interact directly by co-staining the Notch ligand in ECs and receptor in hNSPCs, and demonstrate the spatial distribution of these ligand-receptor pairs.

Response: We have provided evidence for direct interaction between hNSPCs (Notch) and hECs (Notch ligand) in the following ways. Firstly, new data in Figures 2 and Supplementary Figure 4 help to establish direct contact between hNSPC and hEC plasma membranes. Secondly, the scRNAseq data in the paper show up-regulation of hNSPC Notch signaling after hEC co-culture and increases in downstream Notch signaling targets Hes and Hey (Figure 6 and Supplementary Figure 12). Thirdly, disruption of Notch in hNSPCs by DAPT treatment (Figure 6) or reduction of DLL4 in hECs by siRNA (Figure 7) abrogate the ability of hECs to increase type B hNSPCs, providing the best direct evidence of the importance of Notch signaling to the effect. Staining of Notch on hNSPCs in the vicinity of DLL4 on hECs would not prove that the two are binding and leading to signaling in hNSPCs. Thus, data in the paper showing upregulation of Notch pathway genes in hNSPCs after co-culture, and loss of the hEC effect on type B cells after disrupting Notch or DLL4, more directly indicate functional interaction of Notch and DLL4 and resultant downstream Notch signaling in hNSPCs relevant for type B formation.

(4) Additional control experiments (e.g. fibroblasts co-culture) are needed to show that the effect of hECs on hNSPC proliferation is not simply because of the cell density in the co-culture condition.

Figure 5f. showed that in co-culture condition, the total number of hNSPCs is less than control, suggesting less proliferation when co-cultured with ECs. However, this could be simply because there are more cells in this condition (vs. control condition) which make it crowded for cells to grow. Additional control experiments (e.g. fibroblasts co-culture) are needed to show that this effect is not simply because of crowd but specific to endothelial effect.

Response: We have included data from new experiments in which hNSPCs were co-cultured with normal human lung fibroblasts (NHLFs) as controls to address this concern. Analysis of proliferation by EdU incorporation shows no difference in hNSPC proliferation when co-cultured with fibroblasts. The new data is now included in Supplementary Figure 11e.

(5) In Figure 8a, b, the staining of GFAP, Lex, S100A6 is essentially everywhere, this type of staining is not very useful to tell their relationship with blood vessels.

Figure 8a, b, the staining of GFAP, Lex, S100A6 is essentially everywhere, this type of staining is not very useful to tell their relationship with blood vessels.

Response: The human subventricular zone (SVZ) has a fairly high cell density, and the immunohistochemistry data show robust staining of type B cell markers in this region. The fact that staining for these markers is concentrated in the SVZ helps to establish them as valid markers for type B cells *in vivo*.

We added several new pieces of data to deepen the analysis of type B marker expression in the human SVZ. Firstly, we stained human SVZ with antibodies to GFAP and SOX2 and identified cells expressing these two type B cell markers in the SVZ. These data are now included in Figure 8a and Supplementary Figure 16. Secondly, for all type B cell markers analyzed *in vivo* (GFAP, SOX2, S100A6, LeX, and PROM1), we obtained high magnification and high resolution images that allow us to assess cellular co-localization of markers in tissue. New images in Figures 8, 9, and 10 and Supplementary Figures 15-21 provide more detail of staining patterns and show cells with co-localized type B cell markers in multiple samples of human brain tissue at a variety of ages (biological replicates), which speaks to the robustness of the findings. Lastly, confocal imaging of human tissue shows cells expressing type B cell markers (GFAP, SOX2, and PROM1) in contact with CD31-expressing vasculature (shown in Figure 10a-c and Supplementary Movies 1-3).

Reviewer 2

(A) -The authors report an increase in GFAP+SOX2 cells upon hEC co-culture in Figure 1. It is however unclear what the authors mean with % GFAP+SOX2+ cells in Figure 1. Is this amount normalized over the number of SOX2+ cells per condition (1b, 1c, 1d, 1e)? This is need to accurately assess the extent of increase of GFAP+SOX2+ cells as the amount of SOX2+ cells increases as well (1b).

I furthermore wonder about the classification of marker positivity across figure 1. For example, in panel 1c the main difference between media control and +hECs seems to be the intensity of GFAP expression rather than individual cell positivity. The representative image shown does not indicate 20% GFAP+SOX2+ in MC. More representative lower magnification images for all conditions would be useful to be able to evaluate the robustness of this phenotype. Panels in Supplementary Figure 1d should be overlaid with SOX2.

The initial amount and percentage of GFAP+SOX2+ cells should likewise be shown at day 0 (Supplementary 1d and 1e) to understand the starting point. Could the authors comment why there is already a high percentage of GFAP+SOX2+ cells in the media control condition at day 1? Do these data not rather indicate a maintenance of GFAP+SOX2+ cells (Supplementary 1d) rather than increasing GFAP+SOX2+ cells induced by hECs?

[Addressed in responses #1-#4]

(1) The GFAP+SOX2+ cells should be normalized over the number of SOX2+ cells per condition to accurately assess the extent of increase of GFAP+SOX2+ cells as the amount of SOX2+ cells increases as well.

Response: New graphs are now included in Figure 1b, 1c, 1d, and 1e and Supplementary Figure 5 showing GFAP+SOX2+ cells out of SOX2+. The normalized data show an increase in GFAP+SOX2+ cells after co-culture with hECs and hBMECs but not after co-culture with control normal human lung fibroblasts.

(2) More representative lower magnification images for all conditions would be useful to evaluate the robustness of the phenotype.

Response: Lower magnification images for all conditions are now included in Figure 2, a new Supplementary Figure 2, a new Supplementary Figure 3, and Supplementary Figure 5.

(3) Panels in Supplementary Figure 1d should be overlaid with SOX2.

Response: Unfortunately, we do not have SOX2 stains for the images in Supplementary Figure 1d. However, there are multiple examples of SOX2 stains for hNSPCs on top of (Figures 1a, 1d) or adjacent to (Figure 1c, Supplementary Figure 5a) hECs throughout the manuscript. These show high SOX2 expression in all cases.

(4) The initial amount and percentage of GFAP+SOX2+ cells should be shown at day 0 (Supplementary 1e) to understand the starting point. Do these data not rather indicate a maintenance of GFAP+SOX2+ cells rather than induction of GFAP+SOX2+ cells by hECs?

Response: The graph in Supplementary Figure 1e shows across three time points that co-culture with hECs results in a higher percentage of type B cells compared to hNSPCs in mono-culture. In these experiments, cells are settling and adhering on day 0, precluding analysis at that time point. Regarding the question of maintenance compared to induction, we have revised the text throughout to better reflect the data, which shows an increase in type B cells in co-culture compared to media control.

(B) *-Figure 2: More representative and lower magnification images as well as single channel and overlaid panels of the same areas should be shown to evaluate the presence of LeX, GFAP and double positive cells across conditions. Also here, I wonder about the classification of marker positivity, given that it is difficult to see how the quantified stark increase in the GFAP+LeX+ cells matches the representative images. The same holds true for the S100A6 staining. These data (including quantifications) should be repeated with SC30 hNSPCs to increase robustness of the observations.*

[Addressed in responses #5-#6]

(5) Single channel and overlaid panels of the same areas should be shown to evaluate colocalization of LeX and S100A6 with GFAP across conditions.

Response: Single channel images of LeX, S100A6, and GFAP are now included in Figure 2 and Supplementary Figure 3.

(6) The LeX and S100A6 data should be repeated with another set of hNSPCs to increase robustness of the observations.

Response: We repeated co-culture experiments with another set of hNSPCs, SC23, and co-stained with type B cell markers. The new data shows that these cells also robustly express GFAP, SOX2, S100A6, and PROM1 after co-culture. The new data are included in Supplementary Figure 2b, Supplementary Figure 3, and Figure 9.

(C) -As a general note to most quantifications in the paper: what is meant with wording like “5 areas analyzed”. How large are these areas? How were they chosen? Can the authors color the dots belonging to the same biological experiments in the quantitative graphs so that it is easier to understand the data robustness?

[Addressed in responses #7-#8]

(7) For quantitation, what is meant with wording like “5 areas analyzed”. How large are these areas? How were they chosen?

Response: The description of the imaging and region selection are detailed in the Experimental Procedures section of the manuscript. We revised the text to include the size of the imaged area, which was 750 μm x 563 μm .

(8) Can the authors color the dots belonging to the same biological experiments in the quantitative graphs so that it is easier to understand the data robustness?

Response: Graphs have been revised to show distinct symbols for biological replicates to better indicate data robustness.

(D) -The authors report mixed evidence upon conditioned media experiments (Supplementary Figure 2-3). The authors should show overlaid panels of GFAP and SOX2 in Supplementary Figure 2, at high and low magnification. This data is currently derived from $n=1$ and needs to be repeated across lines and independent experiments and include statistical tests. As per before, normalization of GFAP+SOX2+ over total SOX2+ is also needed. Figure 1c and 1d would likewise strongly benefit from including the conditioned media data from Supplementary Figure 3 to the main figures, to put the data variation across the observed effects in the different experimental settings in a more transparent manner. Could the authors attempt to biologically discuss these differences? Finally, the authors should show consistent effects of the direct hEC co-culture over later passages, since the authors argue that CM are not consistent due to the passaging effect, but the reverse is not shown.

[Addressed in responses #9-#11]

(9) The authors should show overlaid panels of GFAP and SOX2 in Supplementary Figure 2 and normalize GFAP+/SOX2+ over total SOX2.

Response: Supplementary Figure 2 is now Supplementary Figure 5 and includes overlays of GFAP and SOX2 at low and high magnification as well as GFAP+SOX2+ cells out of SOX2+.

(10) Could the authors attempt to biologically discuss the differences in conditioned media effects between the different ECs?

Response: We added text to the Discussion section to discuss these differences. Since the main focus of the study was the contact-mediated effects of hECs on hNSPC phenotype, we did not pursue additional non-contact transwell studies. In the Discussion, we highlight that our data do not rule out effects of secreted factors, particularly *in vivo* where secreted factors may be dynamically regulated.

(11) The authors should show consistent effects of the direct hEC co-culture over later passages.

Response: Data for direct co-culture over passage is included in Supplementary Figure 6. To highlight the comparison, we added new graphs in Supplementary Figure 6e and 6h showing GFAP+SOX2+ hNSPCs (panel e) and GFAP+LeX+ hNSPCs (panel h) from co-culture conditions over passage for comparison with the original conditioned media graphs in these panels.

(E) -Figure 3-4: *The single-cell RNA sequencing data is heavily biased to cells derived from the hNSPC mono-culture and hNSPCs with hEC conditioned media, while very few hNSPC cells are retrieved in the hNSPCs-hEC co-culture which is arguably the most relevant setting (as mentioned by the authors in the legend of Supplementary Fig. 6a). Increasing the numbers of the hNSPCs in the hEC co-culture conditions is however needed to reach more robust conclusions, especially when attempting to infer cell type distributions across datasets. SOX2 and LeX (CD15) should be included in the dot plot in fig. 1d to further understand type B classification. Based on the identification now made of the human type B cells, could the authors now identify this population in available single-cell RNA sequencing datasets of human (fetal) brain? This would greatly strengthen the in vitro observations.*

[Addressed in responses #12-#14]

(12) Concern about the lower numbers of hNSPCs in the co-culture condition affecting cell type distribution across datasets.

Response: For the scRNAseq analysis, merged datasets (media control, conditioned media, and co-culture) were used to identify clusters, which ensures data robustness since the maximum number of cells are considered to create clusters. Because of this, cell type distributions across datasets should not be adversely affected by the number of cells in each condition. Notably, the percentage of type B cells identified in the scRNAseq data across the conditions matches well with the percentages identified by immunostaining (scRNAseq: type B cells 56% co-culture and 30% media control; immunostaining shown in Figure1: type B cells 66% co-culture and 16% media control). This provides independent confirmation of the percentages calculated from the scRNAseq data.

(13) SOX2 and LeX (CD15) should be included in the dot plot in Figure 3d to further understand type B classification.

Response: We modified the dot plot in Figure 3d to include SOX2. We cannot include LeX because it is a glycan (rather than a protein) and there are multiple fucosyltransferases responsible for creating the LeX epitope.

(14) Can we go back to published *in vivo* data sets and now find the type B cells in those?

Response: When we originally submitted the manuscript for publication, there were no published datasets from the adult human SVZ that contained type B cells. However, a new publication in 2024 did include these cells, so we were able to compare their scRNAseq data to

ours. The comparison is now included in Figure 4 and Supplementary Figures 8 and 9 and bolsters the identification of hNSPCs in hEC co-culture as human type B cells.

(F) -Figure 5: Based on which marker staining did the authors identify and quantify (total) hNSPCs in 5c-f? Figure 5c seems to show weakly positive AQP4 cells in the previous EC condition. Does this rather indicate impaired differentiation?

[Addressed in response #15]

(15) Based on which marker staining did the authors identify and quantify (total) hNSPCs in 5c-f? Figure 5c seems to show weakly positive AQP4 cells in the previous EC condition. Does this rather indicate impaired differentiation?

Response: In Figure 5, hECs were identified by positive CD31 staining and remaining cells were hNSPCs (CD31-negative). This information is now included in the Experimental Procedures. We agree that weakly positive AQP4 cells after hEC co-culture may indicate impaired astrocyte differentiation, which fits with scRNAseq data showing reduced astrocyte progenitors (Figure 5a) and differentiation data showing lower astrocyte formation (Figure 5c) after co-culture. The Results text has been revised to raise this point.

(G) -Figure 6-7: The authors postulate an importance of Notch signaling in the type B phenotype upon hEC co-culture and experiments with DAPT corroborate this. To strengthen these observations, experiments also with SC30 hNSPCs would increase data robustness. Furthermore, performing the same DAPT experiments in a conditioned medium set-up would further strengthen the need for direct contact, rather than other effects of DAPT. There seems a general variation in GFAP+SOX2+ across experiments performed. Why is there a marginal increase from 35 to 40% in GFAP+SOX2+ hNSPCs in 7f upon hEC with NT siRNA, while this was increased earlier from 20 to 60% in 1b? This questions the robustness of the siRNA experiments and effects observed, since the hEC co-culture experiment seems to not increase significantly GFAP+SOX2+ already by itself. Since conditioned medium experiments also influenced type B phenotype with varied observations, the authors could further investigate the importance of secretion by e.g. blocking general cell secretion with small molecules. It is important that the authors place the importance of Notch in the right context keeping in mind their observations across conditioned medium and co-culture experiments.

[Addressed in responses #16-#17]

(16) Suggestion to perform the DAPT experiments with another set of hNSPCs and with hEC conditioned media.

Response: We attempted the DAPT experiments with additional hNSPCs, but unfortunately the cells did not respond well to DMSO, which is the diluent for DAPT, and we could not complete these experiments. Treating hNSPCs with DAPT prior to culturing them in hEC conditioned media is unlikely to reveal new information because conditioned media-treated hNSPCs do not show an increase in the type B phenotype. Therefore, it is likely that DAPT treated cells would not differ from controls. Further, DAPT-treated media control hNSPCs are included in Figure 6i, j, and they do not differ from DMSO controls in generation of type B cells. These data show there are no off-target effects of DAPT affecting type B cell formation. Collectively, the data in Figure 6 showing that disruption of Notch signaling in hNSPCs disrupts the formation of type B cells after co-culture and the data in Figure 7 showing that DLL4 siRNA treated hECs lose the ability to increase type B cells directly demonstrate the involvement of Notch ligands and receptors in the hEC-induced increase in type B cells.

(17) Why is there less of an increase in GFAP+, SOX2+ hNSPCs in cells treated with non-targeting siRNA in Figure 7 compared to untreated cells in Figure 1?

Response: While the effect is more modest for hNSPCs treated with non-targeting siRNA, hEC co-culture still significantly increases type B formation ($p < 0.0001$) as shown in Figure 7f. We revised the text to mention the importance of the non-targeting siRNA control for comparison to DLL4 siRNA since the non-targeting siRNA may have somewhat lowered the formation of type B cells.

(H) -Figure 8: The evidence for this cell type *in vivo* as it stands is rather weak. The authors should also stain for SOX2 in the tissue. Why do the authors further suddenly switch to PROM1 for the tissue stainings? Do the *in vitro* cells also show PROM1 positivity in the GFAP+SOX2+ cells? Do additional markers identified from Fig. 3-4 show similar overlap? How do the authors explain the abundant number of PROM1+ cells that are not into contact with CD31+ cells? Finally, could the authors stain for active Notch signals, e.g. DLL4, in relation to the proximity of the CD31+ cells in the tissue to strengthen their putative mechanism?

[Addressed in responses #18-#20]

(18) The authors should also stain for SOX2 in the tissue.

Response: We added several new pieces of data to deepen the analysis of type B marker expression in the human SVZ. Firstly, we stained human SVZ with antibodies to GFAP and SOX2 and identified cells expressing these two type B cell markers in the SVZ. These data are now included in Figure 8a and Supplementary Figure 16. Secondly, for all type B cell markers analyzed *in vivo* (GFAP, SOX2, S100A6, LeX, and PROM1), we obtained high magnification and high resolution images that allow us to assess cellular co-localization of markers in tissue. New images in Figures 8, 9, and 10 and Supplementary Figures 15-21 provide more detail of staining patterns and show cells with co-localized type B cell markers in multiple samples of human brain tissue at a variety of ages (biological replicates), which speaks to the robustness of the findings.

(19) Do the *in vitro* cells also show PROM1 positivity?

Response: We used the PROM1 antibody to stain hNSPCs co-cultured with hECs and found that the cells *in vitro* were also positive for PROM1. Staining for two different sets of hNSPCs (SC23 and SC27) with PROM1 and GFAP is now included in Figure 9a. Regarding PROM1+ cells in tissue that are not in contact with CD31+ cells, determining that a particular cell is not in contact with vessels in human brain tissue is not possible because the sections are only 30 μm thick (need to be this thickness for antibody staining) and thus do not fully encompass vessels extending in 3D through the tissue. This makes analysis of hNSPCs relative to blood vessels challenging, because while we might mark an hNSPC as not in contact with a vessel based on that section, we will be unable to ascertain whether that cell is in fact contacting vessels outside of the plane of that section. In animal models, issues such as these have been addressed with tissue clearing protocols. However, human SVZ tissue is too scarce to allow that type of approach. As described in #18 above, we include new high resolution images of the human SVZ showing contact between type B cells and vasculature *in vivo*.

(20) Can the authors include staining for activated Notch in the tissue?

Response: We attempted to assess Notch activation by purchasing and testing three separate antibodies that should detect cleaved Notch, which is an early step in Notch activation. However, we could not confirm that any of the antibodies detected cleaved Notch in immunostaining, so we were unable to use them to assess activated Notch in our studies.

Reviewer 3

(1) Validate the co-culture findings in a 3D system.

- The co-culture system is in a 2D setting, which does not really reflect the in vivo situation. The findings using this 2D system should be validated using a 3D one, where vessel-like structures and NSCs association to vessels is present.

Response: In order to assess our *in vitro* findings in a 3D system, we co-cultured hNSPCs and hECs in a scaffold mimicking brain mechanical and extracellular matrix properties and stained with type B cell markers GFAP and LeX as well as CD31 for hECs and vessels. As observed in 2D environments, co-culture of hNSPCs with hECs in 3D led to increased co-expression of type B markers. These data are now included in Supplementary Figure 4. Of course, human tissue is the ultimate physiological 3D environment for these cells, so we also added new staining of human brain tissue with type B cell markers. These data are included in Figures 8, 9, and 10 and Supplementary Figures 15-21.

(2) Text for CD31 inside the images is not visible. In addition, please include higher magnification images to show hNSPC-hEC contact.

- Figure 1a-e: the text for CD31 inside the images is not visible, please change it to another location. In addition higher magnification and higher resolution images should be shown to show real EC-NSC contacts.

Response: The text for CD31 has been moved in all relevant images. Regarding contact, we performed new experiments using cell surface markers for hNSPCs and hECs since this would allow us to observe cell perimeters and assess cell contact. We used LeX (CD15), which recognizes a cell surface glyco-epitope, for hNSPCs and CD31, which is platelet endothelial cell adhesion molecule (PECAM1) that is expressed on endothelial cell plasma membranes for hECs. Confocal imaging of the stained cells showed hNSPCs spread over hECs and revealed sites of contact between the two cell types. These new images are now included in Figure 2 of the manuscript.

As a second approach, we co-cultured hNSPCs and hECs in three dimensional (3D) scaffolds mimicking brain mechanical and extracellular matrix properties. We found increased type B marker expression (GFAP and LeX) for hNSPCs co-cultured with hECs in 3D, and confocal imaging showed contact between hNSPCs and hECs. These new data are now included in Supplementary Figure 4.

Lastly, confocal imaging of human tissue shows cells expressing type B cell markers (GFAP, SOX2, and PROM1) in contact with CD31-expressing vasculature. These data are shown in Figure 10a-c and Supplementary Movies 1-3.

(3) EC secreted factors have previously been shown to induce proliferation and differentiation of NSCs, so why is that not the case here?

- Figure 2: other studies have shown that ECs secrete factors (i.e. BDNF, IGF1, etc) that induce proliferation and differentiation of NSCs. From the data presented in figure 5, it seems that in the settings used by the authors, hECs conditioned medium does not lead to increase NSC proliferation compared to medium control. Why is this? Is this system then not reflecting similar conditions are used before in other studies? It would to show that their settings can reproduce previously described mechanisms.

Response: Previous studies assessing EC effects on NSPCs have predominantly been carried out with rodent cells. We clarify this in the Introduction and text in the Introduction sets the stage

for highlighting the differences between rodent and human cells. Rodent and human brains differ in development, structure, and function and there are clear molecular and structural differences between human and rodent NSPC niches (described in paragraph 5, Introduction). Our studies with human cells show that hEC contact regulates formation of adult type B hNSPCs. We address in the Discussion the fact that hEC conditioned media in our study did not affect formation of type B cells, but that there could still be *in vivo* hEC secreted signals that are dynamically regulated or not well-captured *in vitro*. Since we do not directly compare EC CM effects between human and rodent cells in our study, we revised text in the Discussion drawing comparisons between the two.

(4) Figure 2 b and e: the ECs should also be shown via staining for a EC marker. How are the contacts of type B NSCs with ECs?

- Figure 2 b and e: the ECs should also be shown via staining for a EC marker. How are the contacts of type B NSCs with ECs?

Response: The hECs stained with CD31 have been added to the LeX/GFAP and S100A6/GFAP images of Figure 2. Regarding contact between hNSPCs and hECs, we performed new experiments using cell surface markers for hNSPCs and hECs since this would allow us to observe cell perimeters and assess cell contact. We used LeX (CD15), which recognizes a cell surface glyco-epitope, for hNSPCs and CD31, which is platelet endothelial cell adhesion molecule (PECAM1) that is expressed on endothelial cell plasma membranes for hECs. Confocal imaging of the stained cells showed hNSPCs spread over hECs and revealed sites of contact between the two cell types. These new images are now included in Figure 2. As described above, we also show contact in 3D scaffolds and human brain tissue.

(5) Figure 2e: it is very difficult to see the colocalization of S100A6 with GFAP. It is recommended that images for single channels are also shown.

- Figure 2e: it is very difficult to see the colocalization of S100A6 with GFAP. It is recommended that images for single channels are also shown.

Response: Single channel images of LeX, S100A6, and GFAP are now included in Figure 2 and Supplementary Figure 3.

(6) Line 227-231: According to this explanation, there are two clusters of endothelial cells: proliferative and non proliferative endothelia cells. It would be important and relevant for this research to study which kind of interactions are mainly represented between NSC-proliferative EC and which kind of NSC-non-proliferative EC interactions. Which ones lead to typeB NSCs?

- Line 227-231: According to this explanation, there are two clusters of endothelial cells: proliferative and non proliferative endothelia cells. It would be important and relevant for this research to study which kind of interactions are mainly represented between NSC-proliferative EC and which kind of NSC-non-proliferative EC interactions. Which ones lead to typeB NSCs?

Response: The scRNAseq data shows that the majority of hECs are non-proliferating (Figure 3a), and express DLL4 (Figure 7a), which we found important for promoting a type B phenotype in hNSPCs. Notch signaling regulates EC proliferation since blocking DLL4 increases proliferation. Thus, non-proliferative ECs would be expected to have higher DLL4 expression compared to proliferative ECs. The effects of hECs on hNSPCs observed in our study are likely due to the majority of hECs, which are non-proliferating and express higher levels of DLL4. We have clarified this point in the Results.

(7) In lines 271-283, the GFAP is first stated as type B cell specific, then it is stated that GFAP is "well-known" to be expressed by both RGCs and type B cells, then at the end again described as type B specific.

- In lines 271-283, the GFAP is first stated as type B cell specific, then it is stated that GFAP is "well-known" to be expressed by both RGCs and type B cells, then at the end again described as type B specific.

Response: We have carefully revised the text to ensure that we do not mistakenly refer to most of the markers described as "specific" since there is much overlap of markers for cells in the neural lineage. In some cases a marker may be up-regulated in one cell type, but this does not necessarily make it specific for that cell type.

(8) In line 272, NTRK2 is cited as type B specific, as shown in Figure 4b. In Figure 4b it seems that NTRK2 is expressed by both cell types present. Same goes, arguably, for BTG1 and TIAM2.

- In line 272, NTRK2 is cited as type B specific, as shown in Fig4b. On Fig4b it seems that NTRK2 is expressed by both cell types present. Same goes, arguably, for BTG1 and TIAM2.

Response: The text lists NTRK2, BTG1, and TIAM2 as genes highly expressed by both mouse type B cells and the human type B cluster. The data in Figure 4b shows that these genes are more highly expressed by type B cells than RGCs. We revised the graph in Figure 4b to better indicate this.

(9) Figure 4 with the comparison between human type B NSPCs and mouse type B NSPCs seems to be rather isolated from the study. Figure 4d could use some control genes that are known specific mRGC markers.

- Figure 4: this figure with the comparison between human type B NSPCs and mouse type B NSPCs seems to be rather isolated from the study. Would it not make sense to have it earlier to show the differences between both vertebrates? Considering the overall lower expression profile of mRGCs compared to mTypeBs (Fig4b), Fig4d could use some control genes that are known specific mRGC markers.

Response: The primary focus of the comparison in Figure 4 is type B cells compared to radial glial cells. At the time of our initial analysis, the only available datasets for type B cells were from mouse, so we compared these to mouse radial glial cells. This enabled us to identify genes more highly expressed in either type B cells or radial glial cells. We could then compare those genes to those expressed by the human NSPCs in our study to determine whether clusters were more like type B cells or radial glial cells. We found the cluster was much more similar to type B cells. We revised the genes for comparison in Figure 4 to include more RGC markers.

When we originally submitted the manuscript for publication, there were no published datasets from the adult human SVZ that contained type B cells. Subsequently, a new publication in 2024 included datasets from the adult human SVZ that contained type B cells. This allowed us to compare their human scRNAseq data to ours and we found that our *in vitro* human type B cells are more similar to *in vivo* human type B cells than to *in vivo* human radial glia. The comparison is now included in Figure 4 and Supplementary Figures 8 and 9 and bolster the identification of hNSPCs in hEC co-culture as human type B cells. We revised the text in the manuscript to clarify the points of comparison.

(10) Figure 4f does not show any noticeable difference between the 2 subtypes depicted for the genes shown.

- Fig4f does not show any noticeable difference between the 2 subtypes depicted for the genes shown.

Response: Data previously in Figure 4f is now in Supplementary Figure 10, and we revised the genes shown to better highlight differences between the two type B subtypes identified by scRNAseq (Figure 10b). Full data of differential gene expression between subtype1 and subtype 2 are shown in Supplementary Table 5, including significance values.

(11) Figure 4g does not demonstrate that S100A6 demonstrates the "greatest co-expression with GFAP", as no other co-expressions are shown. Also, the colour scale makes it extremely hard to determine the true co-expression of these genes. What units are shown in the co-expression legend?

- Fig4g does not demonstrate, as it is written in line 297 that S100A6 demonstrates the "greatest co-expression with GFAP", as no other co-expressions are shown. Also, the colour scale makes it extremely hard to determine the true co-expression of these genes. What units are shown in the co-expression legend?

Response: Co-expression data previously in Figure 4g and Supplementary Figure 7 are now in Supplementary Figure 10. We changed the coloring of the featureplots in Figure 10c to better highlight co-expression between GFAP and markers enriched in one of the two type B subtypes. Heatmaps accompanying each featureplot range from 1-10 (arbitrary units). A new table in Figure 10d clarifies the percentage of type B hNSPCs expressing the different marker genes and the percentage expressing both GFAP and the marker gene (from the scRNAseq data). This analysis shows that S100A6 is the marker with the highest co-expression with GFAP. We revised the text to clarify the comparison.

(12) Figure 5a: please indicate from where are those pie charts calculated? Are they from the scRNAseq data?

- Figure 5a: please indicate from where are those pie charts calculated? Are they from the scRNAseq data?

Response: Yes, the data in the pie charts are from scRNAseq. This is described in the Experimental Procedures section and we have revised the Results and figure legend to better clarify this point.

(13) An important point is the graph shown in Figure 5e, which claims a significantly higher EdU+ cell population in the media control compared to the co-cultures, but the graph shows very similar data populations with what seem 4 clear outliers in the control group.

- An important point is the graph shown in Fig5e, which claims a significantly higher EdU+ cell population in the media control compared to the co-cultures, but the graph shows very similar data populations with what seem 4 clear outliers in the control group.

Response: We agree that in the original version of the graph there were 4 data points in the control group that were higher than the others. However, we ran an outlier analysis and those points were not identified. If we remove those points from the analysis, the data are still significantly different (p value with all data included: $p=0.0009$, p value with 4 points removed: $p=0.0041$). However, we had no rationale for removing those points and instead repeated the experiment to gather additional data. The new data is included in Figure 5e (now $n=4$ for that graph), and the values are significantly different ($p=0.0076$).

(14) Figures 6b and 6e: no axis and units shown for the bar graphs.

- Fig6b and Fig6e: no axis and units shown for the bar graphs.

Response: Enrichr, which was used to generate the data, does not include an axis for bar graphs. The longer and lighter the bar, the higher the significance for that term. This has been added to the figure legend to clarify. The adjusted p-values are included in Supplementary Table 7 and have been added to the figure legend for the significant terms.

(15) Figure 6c seems to be column-normalised for the average expression, so the true difference between the 2 conditions can not be estimated.

- Fig6c seems to be column-normalised for the average expression, so the true difference between the 2 conditions can not be estimated.

Response: We apologize for the oversight and have removed Figure 6c as this dot plot had less than 5 groups, which may lead to misleading representation of data. The remaining dot plots have more than 5 groups.

(16) Figure 6d shows the "# adherent hNSPCs after dissociation (relative)". Since the values are fractions, then these are not the number of cells as the "number sign" (#) would suggest. What is it relative to?

- Fig6d shows the "# adherent hNSPCs after dissociation (relative)". Since the values are fractions, then these are not the number of cells as the "number sign" (#) would suggest. What is it relative to?

Response: We thank the reviewer for noticing the mislabeled axis. We revised the graph, which now shows % adherent hNSPCs after dissociation. We also revised the Experimental Procedures to better explain the experimental design.

(17) Figure 6j: the text for CD31 inside the images is not visible, please change it to another location. In addition higher magnification and higher resolution images should be shown to show real EC-NSC contacts.

- Figure 6j: the text for CD31 inside the images is not visible, please change it to another location. In addition higher magnification and higher resolution images should be shown to show real EC-NSC contacts.

Response (similar to comment #2 response): The text for CD31 has been moved in all relevant images. Regarding contact, we performed new experiments using cell surface markers for hNSPCs and hECs since this would allow us to observe cell perimeters and assess cell contact. We used LeX (CD15), which recognizes a cell surface glyco-epitope, for hNSPCs and CD31, which is platelet endothelial cell adhesion molecule (PECAM1) that is expressed on endothelial cell plasma membranes for hECs. Confocal imaging of the stained cells showed hNSPCs spread over hECs and revealed sites of contact between the two cell types. These new images are now included in Figure 2 of the manuscript. We also show contact between hNSPCs and hECs in 3D scaffolds (Supplementary Figure 4) and in human brain tissue (Figure 10a-c and Supplementary Movies 1-3).

(18) Figure 6k : x axis legends for hEC are not clear, which one is DMSO and which one DAPT?

- Fig6k : x axis legends for hEC are not clear, which one is DMSO and which one DAPT?

Response: We revised the labeling of graphs in the DAPT experiments shown in Figure 6j (was Figure 6k) to improve clarity.

(19) Figure 8: Here the authors use human tissue to show that type B cells contact blood vessels and that they express LeX, S100A6 and PROM1. The quality of the images and the resolution of them does not allow to state their claims.

- Figure 8: Here the authors use human tissue to show that type B cells contact blood vessels and that they express LeX, S100A6 and PROM1. The quality of the images and the resolution of them does not allow to state their claims.

Response: We added several new pieces of data to deepen the analysis of type B marker expression in the human SVZ. Firstly, we stained human SVZ with antibodies to GFAP and SOX2 and identified cells expressing these two type B cell markers in the SVZ. Confocal images are now included in Figure 8a and Supplementary Figure 16. Secondly, for all type B cell markers analyzed *in vivo* (GFAP, SOX2, S100A6, LeX, and PROM1), we obtained high magnification and high resolution images that allow us to assess cellular co-localization of markers in tissue. New images in Figures 8, 9, and 10 and Supplementary Figures 15-21 provide more detail of staining patterns and show cells with co-localized type B cell markers in multiple samples of human brain tissue at a variety of ages (biological replicates), which speaks to the robustness of the findings. Lastly, confocal imaging of human tissue shows cells expressing type B cell markers (GFAP, SOX2, and PROM1) in contact with CD31-expressing vasculature (shown in Figure 10a-c and Supplementary Movies 1-3). The additional images have strengthened the data showing expression of type B cell markers in the human SVZ.

Response to Reviewer

Reviewer 2:

The authors have appropriately addressed most of my concerns. While in the revised version the authors have not increased the numbers of hNSPC cells from the EC-coculture in the single-cell RNA sequencing, which I do consider a shortcoming, I would advise the authors to transparently show this in the main body of the paper, regarding the amount of cells retrieved per condition (i.e. including S11 a-b to the main figure 3) with a brief remark on this in the main text. Of note, the paper has considerably increased in length and amount of figures, which may need to be reconsidered, e.g. Fig.9 and 10 can be merged, and parts of Figure 4 can go into the supplement.

Response:

We increased transparency regarding the number of hNSPCs in the co-culture scRNAseq data by including the number of cells in each condition for scRNAseq in the Results. We also included new text in the Discussion regarding the relatively low number of hNSPCs in the co-culture scRNAseq data as a limitation of the study and reiterated that scRNAseq results were validated with immunostaining and functional assays. Thus, the conclusions of the study do not rely solely on scRNAseq data.

Regarding the figures, we found it too difficult to merge Figures 9 and 10 as it made the immunostained panels very difficult to see. To note, we are not above the journal's maximum number of figures. We slightly reduced the number of plots in Figure 4, but believe the ones remaining are important to the evaluation of the data since too few markers for each cell type can be misleading.